# Identifying teleconnections and multidecadal variability of East Asian surface temperature during the last millennium in CMIP5 simulations

Satyaban B. Ratna[1], Timothy J. Osborn[1], Manoj Joshi[1], Bao Yang[2], Jianglin Wang[2]

[1]Climatic Research Unit, School of Environmental Sciences, University of East Anglia, Norwich, NR2 2BP, United Kingdom
[2]Key Laboratory of Desert and Desertification, Northwest Institute of Eco-Environment and Resources, Chinese Academy of Sciences, Lanzhou, 730000, China

*Correspondence to*: Satyaban B. Ratna (s.bishoyi-ratna@uea.ac.uk)

**Abstract.** We examine the relationships in models and reconstructions between the multidecadal variability of surface temperature in East Asia and two extratropical modes of variability: the Atlantic Multidecadal Oscillation (AMO) and the Pacific Decadal oscillation (PDO). We analyze the spatial, temporal and spectral characteristics of the climate modes in Last Millennium, Historical and pre-industrial control simulations of seven CMIP5/PMIP3 GCMs, to assess the relative influences of external forcing and unforced variability. These models produce PDO and AMO variability with realistic spatial patterns but widely varying spectral characteristics. AMO internal variability significantly influences East Asia temperature in five models (MPI, HadCM3, MRI, IPSL and CSIRO), but has a weak influence in the other two (BCC and CCSM4). In most models, external forcing greatly strengthens these statistical associations and hence the apparent teleconnection with the AMO. PDO internal variability strongly influences East Asian temperature in two out of seven models, but external forcing makes this apparent teleconnection much weaker. This indicates that the AMO-East Asian temperature relationship is partly driven by external forcing whereas the PDO-temperature relationship is largely from internal variability within the climate system. Our findings suggest that external forcing confounds attempts to diagnose the teleconnections of internal multidecadal variability. Using AMO and PDO indices that represent internal variability more closely and minimising the influence of external forcing on East Asia temperature can partly ameliorate this confounding effect. Nevertheless, these approaches still yield differences between the forced and control simulations and they cannot always be applied to paleoclimate reconstructions. So we recommend caution when interpreting teleconnections diagnosed from reconstructions that contain both forced and internal variations.

## 1 Introduction

Coupled ocean-atmosphere processes cause climate variations on interannual to multidecadal timescales (Dai et al., 2015 and Steinman et al., 2015), resulting in persistent temperature and hydroclimate anomalies over both nearby and remote regions (Wang et al. 2017; Coats and Smerdon, 2017), potentially having both immediate and long lasting consequences for society (Büntgen et al., 2011). Assessing teleconnections between ocean and land can be done using a number of methods, that each have their own limitations. The usefulness of the observational record for understanding multidecadal teleconnections is

limited by its length. Paleoclimate reconstructions can provide information on longer time scales, and can also place the current climate regime in a long-term perspective. Several reconstructions of modes of climatic variability such as the Atlantic Multidecadal Oscillation (AMO) and the Pacific Decadal Oscillation (PDO) have been attempted using networks of proxy data, including tree-rings, ice cores, speleothems, coral growth, lake sediments, and documentary evidence (e.g. MacDonald and Case, 2005, Mann et al, 2009, Wang et al, 2017, Fang et al. 2018, Wang et al. 2018). However, limitations in the geographic and temporal coverage of the proxy records, including terrestrial and marine locations, and differing climatic and seasonal sensitivities, affect the ability of these reconstructions to fully represent decadal to centennial variability (Jones et al, 2009; Christiansen and Ljungqvist, 2017; Smerdon and Pollack, 2016; Jones and Mann, 2004; Frank et al., 2010 ).

Global climate model (GCM) simulations can offer complementary long-term perspectives on the behaviour of important modes of climate variability (Atwood et al, 2016; Fleming and Anchukaitis, 2016; Landrum et al, 2013). Such models can also be used to identify the extent of large scale teleconnections between these modes and regional climate (Coats et al., 2013). GCMs also provide a means of separating changes associated with external forcing from those arising by internal variability since they can be run using differing boundary conditions (Schurer et al., 2013, 2014). GCMs with more extensive representation of processes within the climate system also permit more detailed examination of spatial and temporal variations during the last millennium. However, similar to proxy based records, there are also limitations in paleoclimate simulations. Laepple and Huybers (2014) found potential deficiencies in Coupled Model Intercomparison Project Phase 5 (CMIP5) SST variability, with model simulations diverging from a multiproxy estimate of SST variability (that is consistent between proxy types and with instrumental estimates) toward longer timescales. Parsons et al. (2017) found very different pictures (in terms of the magnitude and spatial consistency) of natural variability between the CMIP5 models, including in the North Atlantic, and between models and paleoclimate data in the tropics.

The focus of our study is on the annual mean temperature of East Asia on multidecadal and longer timescales. On these timescales, variability associated with the AMO (Schlesinger and Ramankutty, 1994; Kerr, 2000; Delworth and Mann, 2000; Wang et al. 2017) and the PDO (Mantua et al. 1997; Newman et al., 2016; Buckley et al., 2019) can exert an important influence on the climate over Asia (Qian et al. 2014; Wang et al. 2013, 2018; Li et al. 2017; Fang et al. 2019). Our aims are to: (1) identify the key teleconnections between the AMO, PDO and East Asian temperature in climate models; (2) to determine to what extent external forcings affect these simulated teleconnections; (3) contrast the simulated and reconstructed behaviour of East Asian temperatures on multidecadal timescales; (4) develop recommendations for making unbiased comparisons between model output and paleoclimate reconstructions. We also provide insight into the long-term simulated behaviour of these modes of climate variability with respect to external forcing and internal variability.

The rest of the article is organised as follows: Section 2 describes the climate models and paleoclimate data used in this study, and the methods to calculate the climate indices. Section 3 describes the results associated with AMO and its teleconnection with East Asian surface temperature and Section 4 discusses the results associated with the relationship between PDO and East

Asian surface temperature. The role of volcanic and solar forcing on these aspects of climate variability is demonstrated in Section 5. In Section 6, further results are discussed and conclusions are summarised.

## 2 Data and Methods

### 2.1 Climate model simulations

We select models from the Coupled Model Intercomparison Project Phase 5 (CMIP5; Taylor et al. 2012) that had provided output for all three experiments considered here: (i) pre-industrial control (PI) run with constant external forcing, (ii) Last Millennium (LM) and (iii) Historical experiments with time-varying external forcings. Of the ten GCMs that met this criterion, three models (MIROC, FGOALS, GISS) were excluded because they show a strong drift in the LM or PI simulations (Atwood et al. 2016, Fleming and Anchukaitis, 2016). Atwood et al (2016) found long-term drift in global mean surface air temperature

in MIROC and FGOALS PI simulations, while Fleming and Anchukaitis (2016) found drift in GISS LM simulations during the initial several centuries and excluded these from their PDO analysis. Details of the remaining seven GCMs are summarised in Table 1 including the volcanic and solar forcings used. The forcing and boundary conditions for the LM simulations follow the protocols of Paleoclimate Model Intercomparison Project Phase 3 (PMIP3) as discussed by Schmidt et al. (2011) and Schmidt et al. (2012). The forcings are composed of volcanic aerosols, solar radiation, orbital variations, greenhouse gas ($CH_4$,

$CO_2$, and $N_2O$) concentrations, and anthropogenic land-use changes over the period 850-1849. The Historical simulations are forced with natural and anthropogenic forcing over the period 1850-2000. The comparison of these 'forced' simulations with 'unforced' control simulations provide a means of assessing what portion of the climate variability is attributed to external forcing and what portion of the climate variability reflects purely internal variability. Also, these simulations are useful in providing a longer term perspective for detection and attribution studies.

We interpolate output from all the models to the CCSM4 grid resolution to facilitate intercomparison. We note that the available model simulations were not necessarily continuous from their LM simulations into their Historical simulations, so modes of variability cannot be calculated across 1850. Each model version was the same across all the simulations. Since our focus is on natural variability arising from internal and external causes, we have minimised the influence of any residual long-term drift or of anthropogenic transient forcings by first detrending (removing the linear trend) the LM (850-1849) and

Historical (1850-2000) time series separately and then merging them into a continuous detrended timeseries for the period 850-2000 (LMH hereafter). Our results are not sensitive to the linear detrending of the Historical simulations (see Supplement Figs S1 and S2).

## 2.2 Diagnosing Atlantic and Pacific variability

The AMO and PDO timeseries are diagnosed using the same methods for both the model simulations and the instrumental observations. For the latter we used HadISST (Rayner et al, 2003) and ERSST (Smith and Reynolds, 2004) for the periods 1871-2000 and 1854-2000, respectively.

The AMO index is calculated from the area-weighted North Atlantic (80– 0°W, 0–65°N) monthly mean SST anomaly for the LMH and PI simulations. When the value of the index is positive (negative), it is known as the warm (cold) phase of the AMO. We have considered here two sets of AMO indices, both with (hereafter, AMOr) and without (hereafter, AMOnr) first subtracting the global mean SST anomaly time series (Trenberth and Shea, 2006) from the spatially averaged (North Atlantic) time series anomalies at each time interval. Atlantic SST will exhibit both internal variability and the response to external

forcings (Wang et al. 2017): by subtracting the global-mean SST anomaly, the AMOr index will reflect more closely the variability that is focussed on the North Atlantic region and not the signal of external forcing present in the global SST pattern. The annual mean of the AMO indices generated by the above two methods are regressed over the North Atlantic annual mean SSTs (Fig. 1) to shows the spatial pattern of the AMO; they closely agree with the observations (Fig 1, bottom panel). The simulated behaviour of the AMO in CMIP3 and CMIP5 models is also discussed by Flato et al (2013).

The PDO index is calculated as the leading mode from an empirical orthogonal function (EOF) analysis of annual SST anomalies in the north Pacific region 20-65°N and 110°E-110°W, which are calculated by first removing the long-term monthly means, then subtracting the monthly mean global SST anomaly at each time interval and then forming annual means. The methodology used to calculate PDO is similar to the methods discussed by Mantua et al. (1997), Landrum et al (2013) and Fleming and Anchukaitis (2016). The EOF analysis yields spatial patterns (loadings) and temporal scores (time series). In

its positive (negative) phase, SSTs are above (below) normal along the west coast of North America and below (above) average in the central north Pacific (Fig. 2). The simulated PDO patterns agree well with each other and with the observed pattern (Fig. 2, bottom panel), in terms of the position of strength of the loading maxima and main loading gradients. There are considerable differences in their time series, however, which are discussed later. The ability and limitation of CMIP5 models to simulate PDO characteristics are discussed by Flato et al (2013).

The model simulated AMO and PDO indices are generated using monthly mean SSTs for the LMH (850–2000) and PI simulations, and then converted into annual mean (January-December) values for our analysis. Similarly, for the time series analysis for the temperature over East Asia (TAS), the annual mean (January-December) value is calculated over the land grid points only and area averaged over the region 60-150°E and 10-55°N. We also consider warm (April-September) and cold (October-March) season averages.

## 2.3 Paleoclimate and forcing reconstructions

The GCMs used volcanic forcing from either Gao et al. (2008; hereafter GRA) or Crowley et al. (2008, hereafter CEA) in their last millennium simulations (Table 1). Accordingly, we compare the model and reconstructed data with GRA and CEA, as well as with the newer volcanic forcing reconstruction from Sigl et al. 2015 (hereafter, SIG). Table 1 also lists the solar forcing reconstructions used to drive the GCMs (hereafter VSK, DB and SBF) and we use these to identify the solar signal in the data.

We used proxy-based reconstructions for the AMO, PDO and TAS as summarised in Table 2. We selected these reconstructions instead of alternatives because of the availability of the data for a period that covers at least 1000 years of our main analysis period 850-2000.

 Two AMO reconstructions are used in this study. Mann et al (2009) reconstructed near-global fields of surface temperature using a diverse mix of annual and decadal resolution tree-ring, coral, ice core and sediments records from across the globe. Their AMO series (hereafter AMN09) was computed from the North Atlantic SST grid cells of their reconstructed fields and extends for the period 500-2006 with 10-year low-pass filters. The annually resolved AMO by Wang et al (2017) is based on tree ring, ice core, historical records only from circum-Atlantic land regions and is available for the period 800-2010. Wang et al. (2017) first reconstruct Atlantic Multidecadal Variability (hereafter WN17V) and then subtract an estimate of the externally-forced component (by regressing against solar and volcanic forcing) to obtain a series that represents mostly internal variability, denoted the Atlantic Multidecadal Oscillation (hereafter WN17O). The two reconstructions AMN09 and WN17V provide estimates of the full (both external and internal) variability of the North Atlantic SST. Although Mann et al. (2009) reconstructed near-global SST fields, we have not subtracted the global-mean SST from the Atlantic-mean SST to isolate the *internal* AMO variability (cf. the 'AMOr' series from the models) because prior to 1600 their reconstruction is a linear combination of only two spatial patterns which gives limited information about the Atlantic–global SST difference.

One annual mean PDO reconstruction is based on Mann et al (2009) as described above and hereafter denoted as PMN09. This series is an average of SST grid cells over the central north Pacific region (22.5–57.5°N, 152.5°E–132.5°W): since this region has mostly negative loadings in our EOF-based PDO index (Fig. 2), we multiply PMN09 by minus one to make it comparable to the other PDO indices. Another annually resolved PDO index is from MacDonald and Case (2005; hereafter, MD05) who used only tree ring records from *Pinus flexilis* in California and Alberta to reconstruct PDO for the period 993-1996. The PDO reconstructions, unlike the EOF-based definition in modeled and observed SST datasets, might also contain some signal of external forcing because the proxy records are influenced by externally-forced variability.

The East Asian temperature reconstruction for the warm season is from Wang et al. (2018, hereafter WN18), which uses the mean of seven published reconstructions and is available for the period 850-1999. See Wang et al. (2018) for a discussion of

the underlying reconstructions and their similarity/differences. We repeat some of our analyses with three of the individual reconstructions used in the WN18 composite (see Supplement Fig. S3).

It should be noted that the time sequences of the reconstructed and simulated data are not directly comparable because each will have its own realisation of internal variability. They should, however, be internally consistent so that their teleconnections can be compared on multidecadal time scales, along with any contribution that is externally forced (to the extent that the external forcing matches between the datasets).

## 2.4 Analysis methods

Since our focus is to understand climate variability on multidecadal timescales, all the time series are passed through a 30-year low pass filter using the Lanczos filter. The correlation analyses are tested for statistical significance using the two-tailed student's t-test. The number of degrees of freedom are the number of 30-year long segments minus 2, i.e if low pass filtered 1000-year long time series has 33 independent samples (1000/30) and 31 degrees of freedom. The null hypothesis (that the correlation is zero) is rejected if the correlation value is greater than the corresponding critical value: we test the statistical significance at the 95% level for area-means but lower this to 90% at the grid-cell level because additional noise increases the risk of a type II error.

Spectral analysis of AMO and PDO indices is performed via the Fast Fourier Transform (FFT). In undertaking spectral analysis, we are interested in the overall shape of the spectra, their redness and broad multi-decadal power, and whether these are similar between models, with/without forcing, and between AMO index definitions. We are less interested in the apparent statistical significance of individual periodicities so we do not apply such a statistical test. In order to isolate the near-internal variability from the LMH model runs, in some analyses we regress out the influence of volcanic forcing from the TAS, AMO and PDO timeseries by calculating the regression coefficient of the GRA or CEA volcanic global forcing timeseries against the TAS, AMO and PDO time series, respectively.

## 3. Influences of Atlantic Multidecadal Oscillation

Figure 3 shows the correlation between the AMO index and multidecadal variations in surface temperature in the LMH and PI simulations. The correlation with *area-averaged* East Asia surface temperature (TAS) is also calculated (Fig. 4). In the PI simulations with constant external radiative forcing, internal climate variability alone results in correlations between the AMOnr and East Asian temperature that are generally positive (Fig. 3, column 1) – i.e. a warmer North Atlantic Ocean is associated with warmer East Asian temperatures. There is only widespread statistical significance on a grid-cell basis for four models (MPI, HadCM3, IPSL and CSIRO; Fig. 3, column 1), though correlations with area-averaged TAS are quite strong (multi-model mean > 0.5) and statistically significant for all models except BCC and CCSM4. The strength of the correlation is very sensitive to the presence of external forcing, becoming more strongly positive both spatially (Fig. 3g, h, i) and on an

area-averaged basis (Fig. 4) in the LMH run for all GCMs (all are significant and the multi-model model correlation reaches 0.75). In many models, the strong positive correlation occur mostly because natural external (e.g., volcanic eruptions) forcings cause concurrent warming or cooling in both the Atlantic and East Asian regions (Fig. 7). Indeed, including external forcings changes CCSM4 from having the weakest (and non-significant) correlation to having the strongest (0.84) correlation between the AMOnr and E Asian temperature.

When we remove the global SST anomaly before calculating the AMO index, its association with TAS becomes much weaker. The reduction in correlation is especially large for LMH runs, both on a grid-cell basis (Fig. 3, column 3 vs. 4), and for area-averaged TAS (Fig. 4, AMOr vs AMOnr). This is because much of the external forcing influence on N Atlantic SST also drives global SST, so subtraction of the global-mean SST anomaly prior to the calculation of the AMO removes this externally-forced variability from the AMO. However, because TAS has a different sensitivity to global forcings, its correlation with AMOr is decreased but remains significant for some models (MPI, MRI and CSIRO). Comparing the correlations with the AMOr index, we see that the correlation is notably weaker in the LMH run than in the PI run for the MPI, HadCM3 and IPSL models (Figs. 3 and 4). This is because the forcings generate a response in the TAS but not in the AMOr (any response is mostly removed because global SST is subtracted from Atlantic SST), weakening their correlation. MRI and CSIRO are notable in that their TAS correlations with AMOr are stronger in the LMH run than in the PI run; this could arise if the amplitude of SST response is greater in the Atlantic than in their global mean, so that AMOr still retains an external forcing signal that then correlates with the external forcing signal in E Asian TAS.

In summary, four models show a clear association between the AMO and East Asian temperatures (especially in the northern half of the region considered here) that arises from internal variability (i.e. in the PI run) that remains significant even if the global SST anomaly is subtracted. This suggests that this mode of internal variability only partly projects onto global-mean SST anomalies. Furthermore, the correlations for all models are strongly affected (increased) by the inclusion of external forcing unless the AMO is defined by subtracting the global-mean SST from the Atlantic SST.

These dependencies of the model correlations on the presence of external forcing and on the calculation of the AMO index are important in the context of interpreting reconstructed data. Suppose we wish to use reconstructed data to answer the question "does the AMO, as a mode of *internal* variability, influence E Asian temperatures on multidecadal timescales?" The reconstructions represent the real world (a situation with external forcings) and some AMO reconstructions (e.g. AMN09 and WN17V) have not isolated internal variability of the N Atlantic SST from externally-forced signals (because, for instance, global-mean SST cannot be subtracted before calculating the AMO if global SST has not been independently reconstructed). This situation is equivalent to column 3 of Fig. 3 (LMH runs with AMOnr indices) and a strong positive correlation might be found between the AMO and E Asian temperatures – but this would *not* establish that the AMO, as a mode of internal climate variability, was strongly influencing E Asian temperatures on multidecadal timescales.

The WN18 E Asian reconstruction represents *warm-season* temperature, so we repeated our model analysis but using both warm and cold season temperatures and obtained results that are closely consistent with those using annual-mean temperatures. The reconstructed AMO series all show positive correlations with the WN18 E Asian temperature reconstruction (Fig. 4). Those representing full AMO variability (WN17V and AMN09) have correlations around 0.4, while the correlation with E Asian temperature falls to 0.24 (which is not significant) for the WN17O series representing only internal AMO variability. None of the reconstructed AMO or E Asian surface temperature series correlate significantly with the equivalent simulated series from the LMH runs, indicating that internal variability and any errors in reconstructed climate and forcings dominate the influence of external forcing, or that model response to forcings is unrealistic.

## 4. Influences of Pacific Decadal Oscillation

Similar to the AMO analysis, Figs 5 and 6 show the correlation between PDO and TAS. The PDO is negatively correlated with TAS in Japan, as expected because cooler SSTs lie adjacent to this region when the PDO index is positive (Fig. 2). These negative correlations extend from Japan across large parts of the north east of our region in four (BCC, CCSM4, IPSL and CSIRO) out of seven models, though they vary widely in strength and significance between models, and are mostly weakened by the inclusion of external forcing (LMH cf. PI in Fig. 5). Most models (for both the PI and LMH runs) show a dipole with mainly negative correlations in the north or centre of our region and positive correlations in the south (e.g. parts of India and Southeast Asia). The predominance of negative correlations across the spatial field means that six out of seven models simulate a negative correlation between area-averaged TAS and PDO, though few are significant because we are averaging across regions with opposite correlations (Fig. 6). There are two key differences between the results with and without external forcing. First, the correlations weaken when external forcing is applied to the models (except for the MPI model where it was already weak). Second, the spread in results is much wider in the absence of forcing (with significant negative correlations for BCC and CCSM4). Together, these results suggest that the internal-variability teleconnection between PDO and E Asian temperature is very model dependent but that external forcing consistently weakens the association. The weakening is likely because the forcing drives additional variability in East Asian temperatures but not in the PDO (because we subtract the global-mean SST anomaly prior to calculating it, and use an EOF definition that depends on SST spatial differences rather than mean SST across the North Pacific, Fig. 2). Using this definition (rather than a simple area-mean SST), the model PDOs do not have strong or consistent responses to forcing in the LMH simulations, in agreement with Landrum et al. (2013) and Fleming and Anchukaitis (2016).

We have also compared the reconstructed PDO and TAS time series (Fig. 6). The correlations are also negative: MD05-WN18 and PMN09-WN18 are -0.10 and -0.41, respectively, but only the latter is significant. The weak correlation with MD05 might be partly related to the fact that this reconstruction is based on only two tree-ring records of North America, suggesting more uncertainty if teleconnection patterns it relies on change through time. The simulated PDO series show very weak correlations with the reconstructed PDO series: as with the AMO, this implies that an external forcing influence is weak compared with

internal variability and reconstruction errors, or that models' PDO response to forcings is unrealistic. For the simulated PDO indices, any external forcing influence may be weak if the PDO calculation (an EOF analysis with the global-mean SST removed) effectively removes a forcing signal. As before, we repeated our model analysis using both warm and cold season E Asian temperatures and found very similar results to the annual temperatures.

## 5. Effect of volcanic and solar forcing

The behaviour of the AMO and PDO timeseries and their correlation with E Asian temperature are clearly sensitive to the presence of external forcing (e.g. LMH versus PI differences), so we now consider the effect of volcanic and solar forcings. Volcanic forcing is the largest external influence within the LM runs (Atwood et al. 2016) and is a key driver of the Little Ice Age and other cooling periods during the last millennium (Briffa et al. 1998; Ammann et al. 2007; Atwood et al. 2016). However, Wang et al. (2018) showed that their E Asian temperature reconstruction has significant multidecadal correlations with solar forcing reconstructions. So, we have compared the model simulations and reconstructions to both volcanic (GRA, SEA and SIG; Fig. 7a) and solar forcing (VSK, DB and SBF; Fig. 7b). All three volcanic forcing timeseries (GRA, CEA and SIG) are closely correlated with each other but are by no means identical (Fig. 7a). For the period A.D. 850-2000, their correlations range from 0.75 to 0.82 (for pairs of GRA, CEA and SIG volcanic forcing) and from 0.80 to 0.91 (for pairs of VSK, DB and SBF solar forcing), highlighting the value in considering multiple forcing histories.

Visually, it is clear to see that the simulated E Asian temperatures and AMOnr timeseries display a multidecadal association with volcanic forcing, while AMOr and PDO timeseries do not (Fig. 7). The colder climate over East Asia and the Atlantic (i.e. AMOnr) is especially evident during the three periods containing strong eruptions (1250s, 1450s and 1810s). Potential influences of solar forcing are harder to identify because the forcing is weaker and shows less distinct episodic behaviour than the volcanic forcing. Turning to correlations (Fig. 8) some key behaviours are clear as discussed below.

First, simulated E Asian temperature is positively correlated with volcanic forcing (significant for all models except BCC) but this is not the case for reconstructed temperatures (on these multidecadal timescales at least; WN18 also report that the volcanic signal is small compared with other influences). We tested to see if the latter result was due to our use of the Wang et al. (2018) composite of reconstructions rather some individual reconstructions that might better resolve the response to volcanoes, but obtained similar results (Supplement Fig. S3). E Asian temperatures are also positively correlated with solar forcing but these are weaker than with volcanic forcing and are either insignificant or marginally significant for all models and reconstructions.

Second, simulated AMOnr timeseries are more strongly (positively) correlated with external forcings than are the AMOr timeseries. With volcanic forcing, the AMOnr correlations are significant for all models except BCC (mean correlations between 0.4 and 0.5 with all three volcanic forcing datasets), while with solar forcing the model correlations are typically

between 0.0 and 0.3. Removing the global-mean SST prior to calculating the AMO (i.e. AMOr) reduces the correlations with both volcanic and solar forcing (Fig. 8). The reconstructed WN17O series is not expected to correlate with external forcings because Wang et al. (2017) removed a regression estimate of the forced signal in reconstructed Atlantic SST to obtain this series. The other two reconstructions (WN17V and AMN09), however, also show only weak correlations with the external forcings.

Third, the simulated and reconstructed PDO timeseries do not correlate significantly with either volcanic or solar forcing (with a couple of exceptions that might be expected by chance – the IPSL model with volcanic forcing and the PMN09 reconstruction with SBF solar forcing). The means of the model correlations are close to zero.

These results explain some of the earlier findings concerning influences on E Asian temperature, specifically that external forcing strengthens its positive relationship with the AMO but weakens its (generally negative) relationship with PDO. Strong positive correlations demonstrate that natural external forcings cause concurrent warming or cooling in both the Atlantic and East Asian regions, contributing to the strengthening of positive correlations between AMOnr and TAS in the LMH simulations compared with the PI runs (Fig. 3 and 4). Removing the global SST anomaly first to obtain the AMOr index renders the correlations with volcanic forcing insignificant (Fig. 8) and the AMOr timeseries (Fig. 7e) do not show any cooling with corresponding volcanic forcing eruptions. This contributes to the much-reduced correlations with TAS when the AMOr series are used (Fig. 3 and 4).

In terms of Pacific variability, the PDO is not strongly correlated with external forcing (as expected because it is diagnosed as an EOF of SST anomalies having first subtracted the global-mean SST). Therefore, adding external forcing does not greatly affect the PDO but it does cause additional variability in E Asian temperature, thus weakening the negative PDO-TAS relationship (Fig. 5 and 6; correlation measures the relative strength of their *common* variability). We might expect this effect to be particularly noticeable in those models that simulate a strong positive correlation between East Asian TAS and volcanic forcing (Fig. 8e). This is the case for CCSM4 (strong volcanic signal in TAS, adding the external forcing greatly weakens the PDO-TAS relationship) but not for MPI (strong volcanic signal in TAS but the PDO-TAS relationship is stronger in LMH than in PI, probably because this model has a negative PDO-volcanic correlation) nor BCC (the PDO-TAS relationship is weaker in LMH than in PI despite there being little influence of volcanic and solar forcings on E Asian temperature in this model).

Since we can diagnose AMO and PDO timeseries that are not significantly correlated with volcanic forcing (Fig. 8: AMOr and the EOF-based PDO), we further tried factoring out the volcanic influence from the TAS time series to see if we could reproduce the behaviour of the control runs (PI) using data from the forced runs (LMH). This is akin to trying to identify the behaviour of internal variability in the real Earth system (Steinman et al., 2015; Dai et al., 2015). We regressed volcanic forcing (using the series for each model – Table 1) on the TAS data, and removed it to yield a TAS series without the linear influence

of the volcanic forcing (see section 2.4). We note that volcanic eruptions may have lagged effects on the oceans from years to decades (e.g. Pausata et al. 2015) and zero-lag regressions may not fully account for this kind of impact on the climate system.

Factoring out volcanic forcing from TAS did weaken the correlations with the simulated AMO LMH series (Fig. 4; AMOnr vs AMOnr_vo for LMH) so that the mean of the model correlations is very close to that found during the PI runs. However the TAS–AMOnr correlations for most individual models differ between PI and the LMH with volcanic influenced factored out. Similar results are found using the AMOr index: there is agreement between PI and LMH correlations for the mean of the model correlations but not for the individual models (e.g. of the four models that show significant positive correlations in the PI runs, only two are significant in the LMH runs with volcanic influence on TAS factored out). Despite factoring out the influence of the dominant forcing and using an AMO index that is not strongly correlated with forcing, we still find very different behaviour in the PI simulation than in the LMH simulation. For two models (BCC and CCSM4), we correctly infer the small role of AMO internal variability on E Asian TAS from the LMH run; for two models (MPI and CSIRO), we correctly infer a significant AMO role (though underestimating its importance for MPI); for two models (HadCM3 and IPSL) we fail to find the significant AMO role; and for MRI we find a significant AMO role despite its PI run showing no significant AMO role on E Asian TAS.

Factoring out the volcanic signal hardly changes the simulated relationships between E Asian TAS and the PDO (PDO_vo; Fig. 6). For some models, therefore, we are not able to determine the internal variability teleconnection between PDO and TAS when external forcings are present despite using a PDO definition that is insensitive to external forcing and regressing out the volcanic influence on TAS. This outcome is especially clear for the two models (BCC and CCSM4) whose internal variability shows strong, significant associations between PDO and E Asian TAS but correlations are weak and insignificant in their LMH runs.

In the BCC model the relationship between volcanic forcing and both AMO and E Asian TAS is notably weaker than the other models (Fig. 8). This behaviour is the same if we use warm or cold-season TAS (not shown) rather than annual-mean TAS. The smaller influence of volcanic forcing for BCC partly explains why it has the weakest correlation between TAS and AMOnr during the LMH simulations. To explore this weak BCC response to volcanic eruptions we analyse net incoming shortwave radiation anomalies composited for the three largest volcanic eruption events in all models (Fig. 9). This shows that the decrease in net incoming shortwave radiation following these eruptions in BCC is less than 25% of the response in both CCSM4 and MPI. The other four models lie between BCC and MPI. We also have analysed the same for the major volcanic events during the historical period for BCC, CCSM4 and MPI but did not find such a weak BCC response compared to the other two models. The weak volcanic forcing and response in BCC may be an artefact of how they implemented volcanic forcing in their last millennium simulation.

To assess the role of external forcing further, the power spectra of the AMO and PDO indices are analysed (Fig.10 and 11, respectively) using the annual mean data (i.e. the data have not been low pass filtered before spectral estimation) for the PI and LMH experiments, reconstructions and instrumental data. All AMO timeseries have red spectra (Fig. 10) at short timescales (up to 20 years) but at multidecadal timescales the redness, the absolute power and the presence of enhanced power across a broad range of frequencies all depend on the model, the presence/absence of forcing and the choice of AMO index. For the AMOnr index, the inclusion of external forcing (LMH runs) greatly strengthens the multidecadal power of the variability; the increase is more moderate for BCC (see earlier discussion). Removing the global-mean SST first (AMOr), reduces the difference between the PI and LMH runs. Some models (BCC, HadCM3 and IPSL) show enhanced power at about 20 years, and there is prominent power around 30 years in the IPSL forced simulation (LMH) that is partly reduced using the AMOr index. In the reconstructions, the spectra have steep gradients over the 40 to 60 year timescales, with elevated power above 60 years. In most models, variability around 60-80 years, often considered typical for the AMO, is quite strong but only MPI has power that is notably enhanced relative to the rest of its spectrum (CCSM4 has elevated power around 40-50 years). The AMO reconstructed spectra provide a comparison (Fig. 10a) of the overall spectral shape. MPI, HadCM3 and IPSL models have the reddest spectra and for AMOnr LMH these are qualitatively similar to the spectra of the WN17V and AMN09 reconstructions, while the other models show much less multi-decadal power.

In the case of the PDO, all models show red spectra with enhanced power at ~15-20 years for both PI and LMH simulations, indicating that the Pacific variability arises mostly by internal variability. Landrum et al. (2013) found the same frequency for both control and last millennium simulations of the CCSM4 model. The enhanced PDO power at ~15-20 years is filtered out by the 30-year smoothing used for the majority of analyses for reconstructions and models reported here, which might weaken the PDO-TAS correlation.

## 6. Discussion and Conclusions

In this study, we identify the key teleconnections between the AMO, PDO and East Asian temperature in climate models on multidecadal time scales and determine to what extent external forcings affect these simulated teleconnections. The instrumental record is too short to clearly distinguish contributions from natural internal variability, natural external forcings and anthropogenic forcings at multidecadal timescales. Zhang et al. (2018) and Wang et al. (2018) have explored this issue using paleoclimate reconstructions of temperatures over a large region in E Asia. Here, we complement these studies by using the dynamical information provided by seven climate models (Table 1), with and without the influence of external forcings over the last millennium. Our key findings are:

1. The models simulate multidecadal modes of variability in the extratropical oceans (AMO and PDO) with spatial patterns similar to those previously identified in the observations and proxy-based reconstructions. Using commonly applied methods to diagnose their time series (area-averaged North Atlantic SST for the AMO and the leading EOF

of North Pacific SST for the PDO) we find that they have spectra with enhanced multi-decadal variability similar to those found in observation and reconstructions (Fig. 10 and Fig. 11). However, the shape and amplitude of their spectra differ between the models and depend on the presence or absence of external forcing. Similar differences were also reported by Parsons et al. (2017) and Fleming and Anchukaitis (2016).

5.  2. These multidecadal modes of variability, along with variations in volcanic forcing, are found to influence E Asian temperature in the models. In most cases, E Asia temperature is positively correlated with the AMO and volcanic forcing, and negatively correlated with the PDO. The correlations are not spatially uniform, with PDO correlations strongest in the parts of the E Asian region that are closest to the extratropical North Pacific Ocean, and the AMO influence showing some latitudinal structure in most models.

10. 3. The presence of external forcing strongly affects the apparent teleconnections between these multidecadal modes of variability and E Asia temperature. The effect depends on how the modes of variability are diagnosed and whether the forcings add common variability to both series (e.g., in the case of AMO) or add distinct variance to E Asia temperature but not to the mode of variability (e.g., in the case of PDO).

4.  If the AMO is defined simply as the mean N Atlantic SST then external forcing strengthens the AMO-E Asia temperature correlation by causing concurrent warming or cooling in both the Atlantic and E Asian regions. For all seven models, the correlations between E Asian temperature and the AMO are stronger in the last millennium forced simulation than in their corresponding control runs when the AMO is defined this way.

5.  Defining the AMO as the difference between N Atlantic and global-mean SST reduces the effect mentioned in point 4 above because part of the externally-forced signal is present in both SST series and the forced signal is reduced by computing their difference. The external signals in N Atlantic and global-mean SST are not identical, however, so using this AMO definition still does not yield correlations between E Asian temperature and the AMO that match those present in the model control runs. We still find different behaviour in the PI simulation than in the LMH simulation even if we also factor out the influence of the dominant forcing (volcanic) on E Asian temperature and use an AMO index that is not strongly correlated with forcing.

25. 6. The PDO definition used here (the leading EOF of Pacific SST minus global-mean SST) yields an index that has only weak correlations with external forcing. Despite this, the multidecadal correlation between the PDO and E Asia temperature (which is negative) is still sensitive to the presence of external forcing. In this case, external forcing *weakens* the apparent teleconnection in two of the models, partly because external (especially volcanic) forcing generates a response in E Asia temperature but not in the PDO index, thus weakening the correlation between them.

Regressing out the influence of volcanic forcing on E Asia temperature has limited effect on the correlation, which remains much weaker than in the control run for two of the models.

These results have significant implications for attempts to determine the influence of the AMO and the PDO strictly as modes of *internal* variability on E Asia temperatures. With models, we can simply analyse their control runs. For the real world, we do not have that option: we can analyse only reconstructions from a real world in which natural external forcings are present. In this case, we recommend, on the basis of our results, that careful consideration be given to separating out the influence of external forcings on both the indices of modes of variability and on the E Asia temperature series before determining the internal variability teleconnections.

There are a number of ways to separate internal and external variability, each with limitations. The modes of variability might be defined using the difference between regional and global SST. For example, the Last Millennium Reanalysis (Tardiff et al., 2019) provides globally-complete temperature fields that could be used (though these are not independent of the CCSM4 climate model or the forcings used). In many cases an independent reconstruction of global SST may not be available. Mann et al. (2009) reconstructed a global field of SST but prior to 1600 their reconstruction is a linear combination of only two spatial patterns which gives limited information about the Atlantic–global SST difference. Another approach is to identify and remove the influence of external forcing, e.g. by regression against forcing histories in reconstructions (Wang et al., 2017, 2018), by a method combining observations with ensemble of coupled climate model simulations (Dai et al., 2015 and Steinman et al., 2015), or using more sophisticated detection and attribution methods (e.g. Hegerl and Zwiers, 2011). These approaches require accurate forcing histories. A further approach is to use an EOF-based definition of the index where the spatial pattern has regions with loadings of opposite sign. External forcing tends to project similarly onto regions with opposite signs, cancelling out much of its influence on the resulting index. This is appropriate for the PDO, as used here, but less so for the AMO because the associated SST pattern is dominated by anomalies of the same sign (Fig. 1). Regression against forcing histories can also be used to remove the influence of external forcing on the target region (E Asian temperature for this study), prior to identifying the influence of AMO and PDO variability (Wang et al., 2018).

Even if these approaches are taken, it may still be impossible to determine the influence of multidecadal internal variability by analysing data that has been subject to external forcings. Despite factoring out the influence of the volcanic forcing in the AMO index, we still find different behaviour in the forced simulation than in the control run for some models. For example, in the forced run of the MPI model, the apparent AMO teleconnection on E Asian temperature is only 0.4 (AMOr_vo in Fig. 4) whereas in the control run it is strong (> 0.7 for AMOnr). For the BCC and CCSM4 models, we can correctly infer from the forced run that AMO internal variability has only a small influence on E Asia temperature provided we define the AMO as the difference between Atlantic and global SST (just factoring out the influence of external forcings from E Asia temperature in insufficient). Similar limitations were found regarding the PDO teleconnection: the PDOs in two models (BCC and CCSM4) have strong negative correlations with E Asia temperature but we cannot correctly diagnose this behaviour if we use a

simulation with external forcings. We note the possibility that external forcing may have modified the dynamical behaviour of the internal variability in these cases, confounding the notion that we can clearly separate forced change from internal variability.

Finally, we found only partial agreement between the behaviours shown by the reconstructions and models. The correlations between E Asia temperature and the AMO and PDO showed the same signs in the models and the data, but the correlation values had a wide range. The strong influence of volcanic forcing in six of the models was not found in the reconstructions (Wang et al. 2018). We need to be careful while interpreting the results of the CMIP5-PMIP3 last millennium simulations in light of the paleoclimate record, because large uncertainties exist in the characterization of volcanic forcing, reconstruction of aerosol loading, optical depth and aerosol effective radius as a function of time, latitude, and height in the atmosphere, all of which exert important controls on the climate system (Atwood et al. 2016).

**Data availability.**

Results for individual Coupled Model Intercomparison Project Phase 5 (CMIP5) models are available for download from the database https://esgf-node.llnl.gov/search/cmip5/ (last access: 15 June 2019). The instrumental SST data HadISST (Rayner et al, 2003) and ERSST (Smith and Reynolds, 2004) are available from https://www.metoffice.gov.uk/hadobs/hadisst/ (last access: 1 June 2019) and https://www.esrl.noaa.gov/psd/data/gridded/data.noaa.ersst.v5.html (last access: 1 June 2019) respectively. Volcanic forcing [Gao et al. (2008), Crowley et al. (2008)] and solar forcing [Vieira et al. (2011), Delaygue and Bard (2011), Steinhilber et al. (2009) and Wang et al. (2005)] data are available as a supplementary data file from https://doi.org/10.5194/gmd-5-185-2012 (last access: 10 February 2018). The other volcanic forcing data by Sigl et al. (2015) are available as supplementary information at https://doi.org/10.1038/nature14565 (last access: 15 November 2018). The AMO and PDO data by Mann et al. (2009) are available as Supporting Online Material from https://doi.org/10.1126/science.1177303 (last access: 10 February 2019). The AMV and AMO data by Wang et al. (2017) are available at https://www.ncdc.noaa.gov/paleo-search/study/22031 (last access: 10 June 2018). The East Asian surface temperature composite data by Wang et al. (2018) are available at https://www.ncdc.noaa.gov/paleo-search/study/27372 (last access: 15 June 2019). The PDO data by MacDonald and Case (2005) are available at https://www.ncdc.noaa.gov/paleo-search/study/6338 (last access: 10 February 2018). The other summer temperature reconstructions for East Asia used in our supplementary material are from Cook et al. (2013) (https://www.ncdc.noaa.gov/paleo/study/19523; last access: 5 January 2019), Shi et al. (2015) (https://www.ncdc.noaa.gov/paleo/study/18635; last access: 5 January 2019) and Zhang et al. (2018) (https://www.ncdc.noaa.gov/paleo-search/study/23491; last access: 5 January 2019).

**Supplementary material.**

[Link to the supplementary material]

**Author contributions.** TJO designed the study. The data analysis was led by SBR with contributions from TJO and MJ. All the figures are made by SBR. BY and JW provided the East Asia temperature reconstruction data and contributed to discussion of the results. All authors contributed to the writing of the paper.

**Competing interests.** The authors declare that they have no conflict of interest

**Acknowledgements.**

Support provided by the Belmont Forum and JPI-Climate project INTEGRATE (An Integrated data-model study of interactions between tropical monsoons and extratropical climate variability and extremes) with funding by UK NERC grant NE/P006809/1. J.W. also acknowledges the support by the National Key R&D Program of China (grant: 2017YFA0603302) and the National Science Foundation of China (NSFC; grant: 41602192). We are thankful to two anonymous referees whose constructive comments helped to improve our study.

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

**Table 1:** Summary of the CMIP5-PMIP3 climate models considered in this study and the volcanic and solar forcing applied in their Last Millennium simulations. The simulation length of last millennium and Historical simulations together for all the seven models are 1151 years (850-2000).

| Model (abbr.) | Institution, Country | Reference | Resolution | | PI length (years) | Forcing | |
| | | | Atmosphere | Ocean | | Volcanic[1] | Solar[2] |
|---|---|---|---|---|---|---|---|
| bcc-csm1-1 (BCC) | Beijing Climate Center, China | Xin et al. (2013) | 128x64 L26 | 360x232 L40 | 500 | GRA | VSK+WLS |
| CCSM4 (CCSM4) | National Center for Atmospheric Research, USA | Landrum et al. (2013) | 288x192 L26 | 320x384 L60 | 1050 | GRA | VSK |
| MPI-ESM-P (MPI) | Max Planck Institute for Meteorology, Germany | Giorgetta et al. (2013) | 196x98 L47 | 256x220 L40 | 1150 | CEA | VSK+WLS |
| HadCM3 (HadCM3) | Met Office Hadley Centre, UK | Schurer et al. (2013) | 96x73 L19 | 288x144 L20 | 1100 | CEA | SBF+WLS |
| MRI-CGCM3 (MRI) | Meteorological Research Institute, Japan | Yukimoto et al (2012) | 320x160 L48 | 364x368 L51 | 500 | GRA | DB+WLS |
| IPSL-CM5A-LR (IPSL) | Institut Pierre Simon Laplace, France | Dufresne et al. (2013) | 96x95 L39 | 182x149 L31 | 1000 | GRA | VSK+WLS |
| CSIRO Mk3L v1.2 (CSIRO) | University of New South Wales, Australia | Phipps et al. (2011, 2012) | 64x56 L18 | 128x112 L21 | 1000 | CEA | SBF |

[1]**Volcanic forcings:** GRA = Gao et al. (2008); CEA = Crowley et al. (2008)

5    [1]**Solar forcings:** VSK = Vieira et al. (2011); DB = Delaygue and Bard (2011); SBF = Steinhilber et al. (2009); WLS = Wang et al (2005)

**Table 2:** Summary of the paleoclimate reconstructions considered in this study

| Reconstruction (abbr.) | Reference | Variable | Time span | Data source |
|---|---|---|---|---|
| AMO (AMN09) | Mann et al. (2009) | N Atlantic average SST | 500-2006 | Tree ring, coral, ice & sediment cores |
| AMO (WN17V and WN17O) | Wang et al. (2017) | N Atlantic (0-70°N) average SST | 800-2010 | Tree ring, ice core, documentary |
| PDO (PMN09) | Mann et al. (2009) | N Pacific (22.5-57.5°N, 152.5°E-132.5°W) average SST | 500-2006 | Tree ring, coral, ice& sediment cores |
| PDO (MD05) | MacDonald & Case (2005) | N Pacific (north of 20°N) principal component of SST | 993-1996 | Tree ring |
| TAS (WN18) | Wang et al. (2018) | E Asia land air temperature | 850-1999 | Mean of 7 available reconstructions |

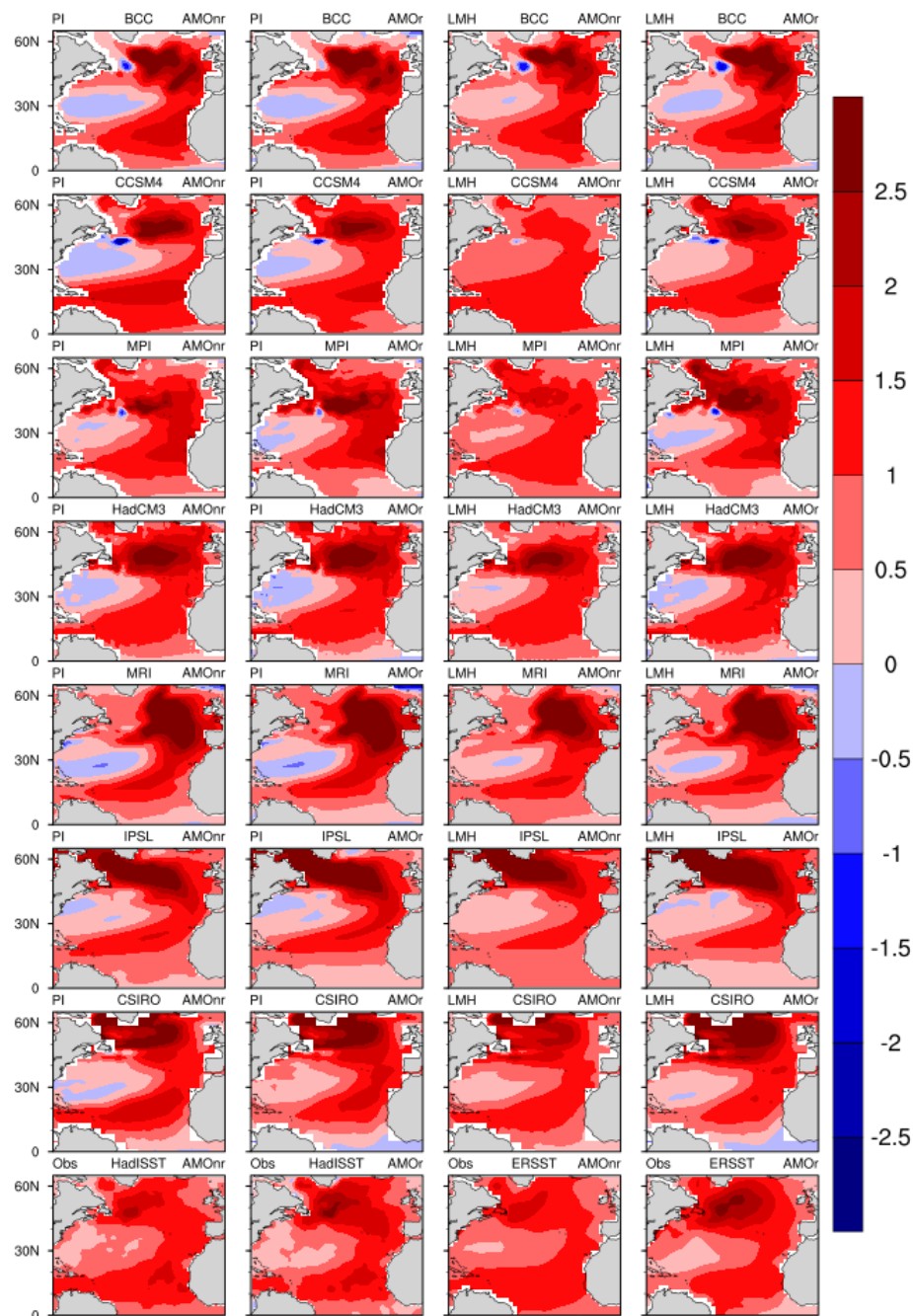

**Figure 1: AMO patterns defined by regression of annual mean SST on the AMO index for each GCM for the PI (1st and 2nd column) and LMH (3rd and 4th columns) experiments compared with HadISST and ERSST observation (bottom row). 1st and 3rd columns use the AMO index calculated without subtracting the global SST anomaly (AMOnr) while 2nd and 4th columns use the AMO index after subtracting the global SST anomaly (AMOr).**

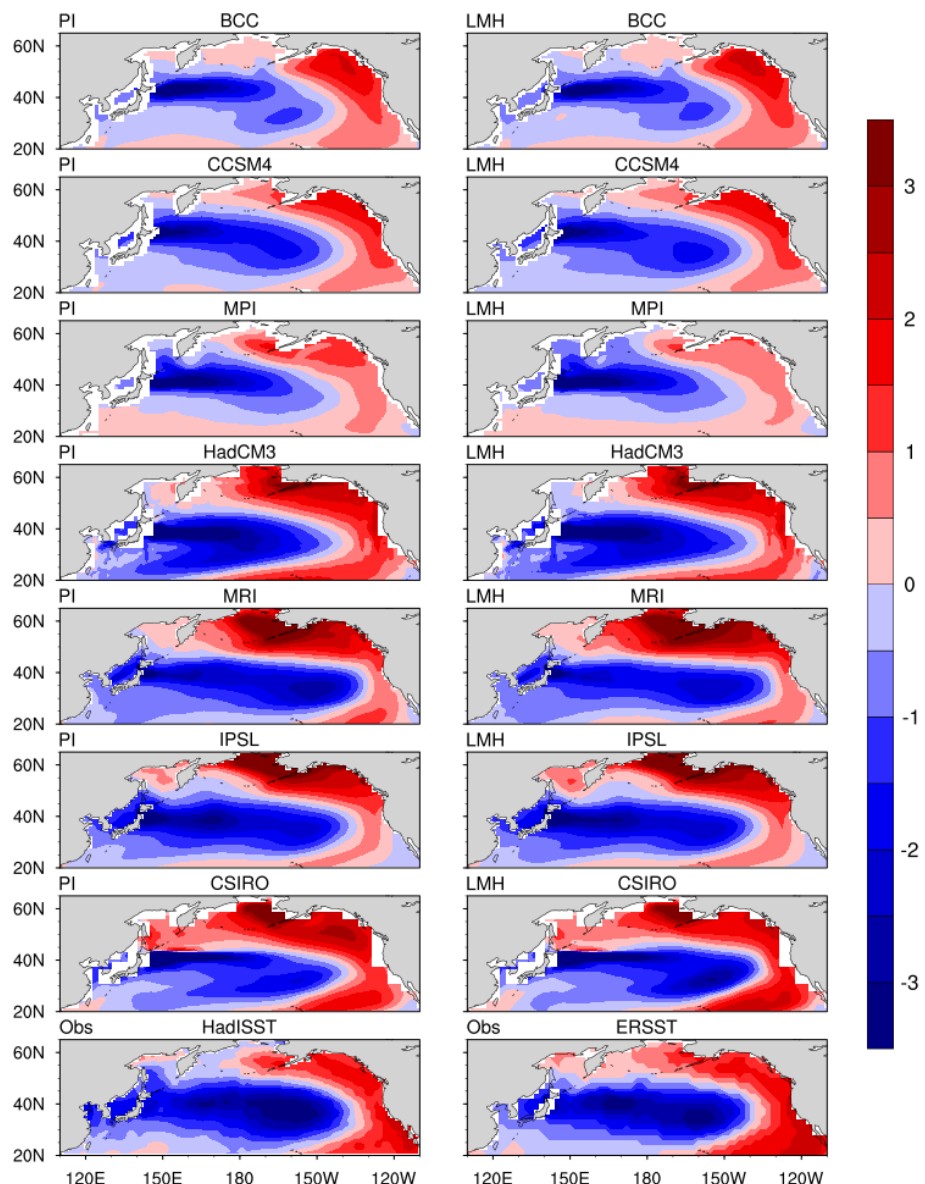

**Figure 2: PDO SST patterns defined by the first EOF of annual mean SST for each GCM for the PI (1st column) and LMH (2nd column) experiments compared with the HadISST and ERSST observations (bottom row).**

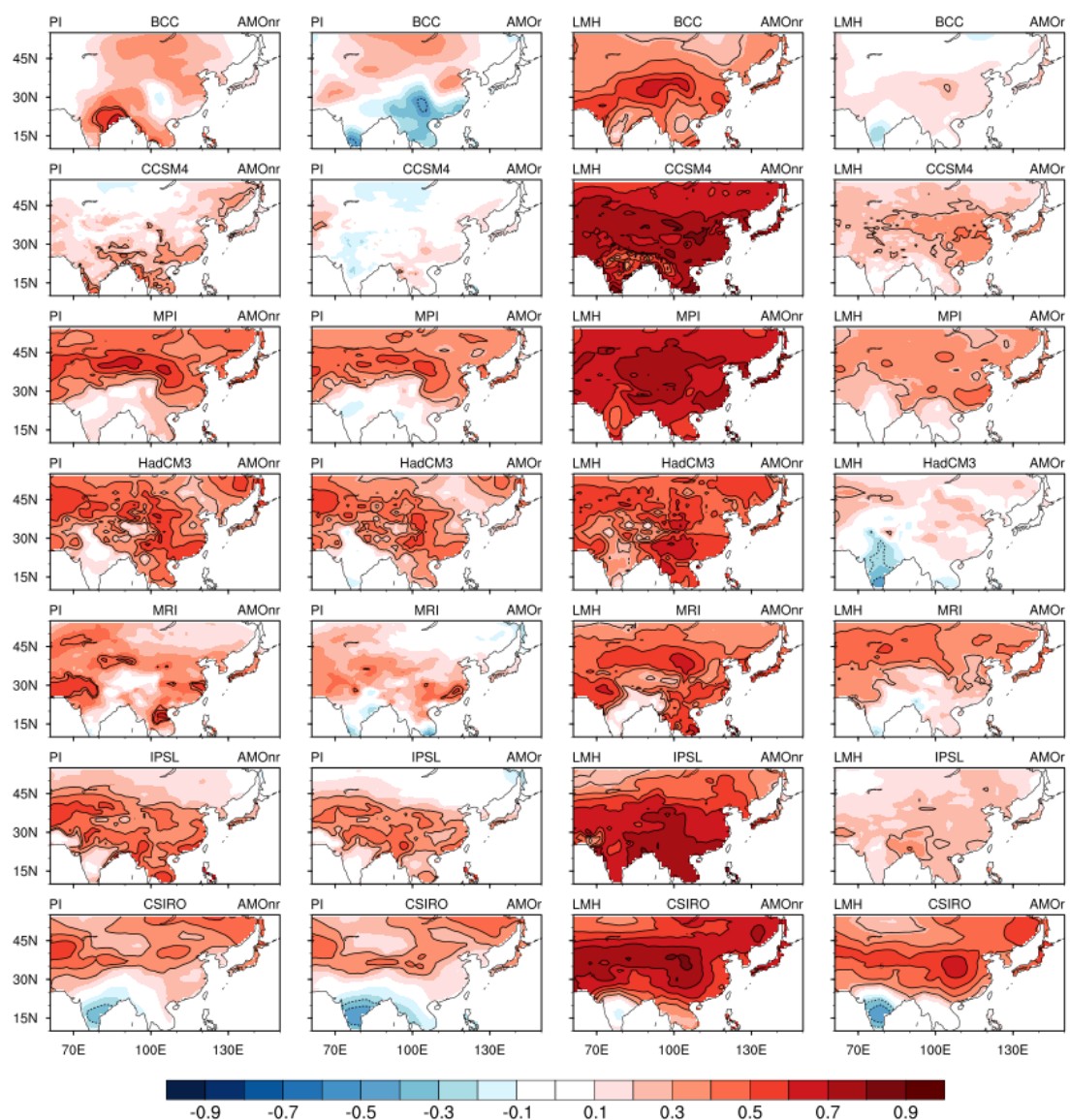

**Figure 3: Correlations between annual mean East Asian surface temperatures and AMO index for each GCM for PI (1st and 2nd column) and LMH (3rd and 4th columns) experiments. AMOnr and AMOr are the two definitions of AMO index as described in Fig 1. Correlation values significant at 90% levels using two-tailed Student's t-test are contoured.**

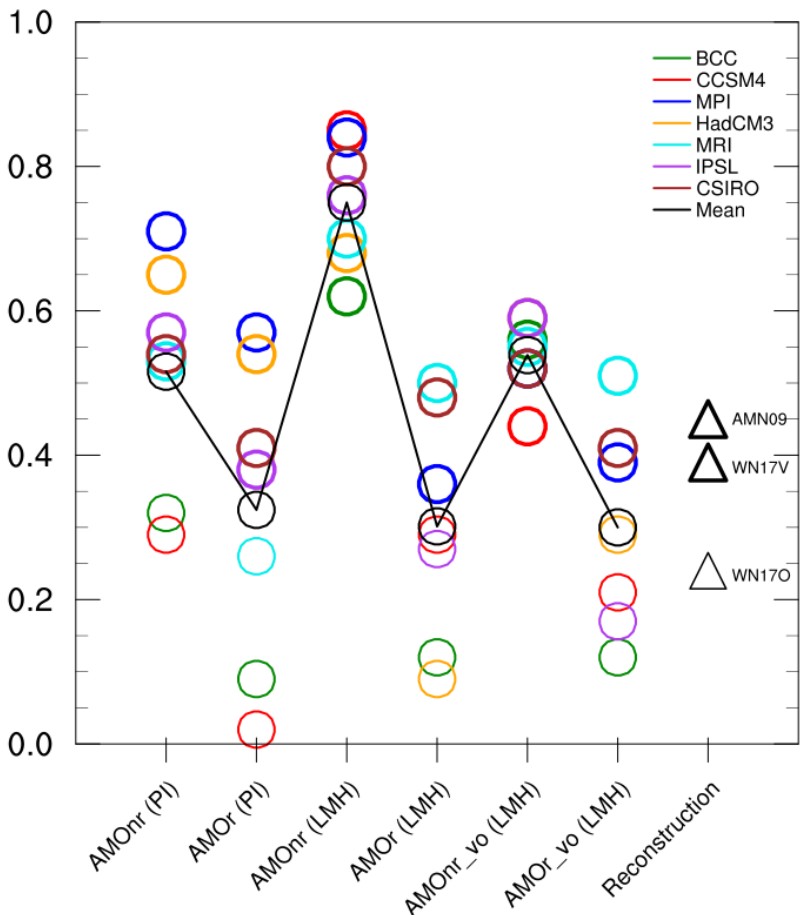

**Figure 4: Correlations of annual mean East Asian regional average surface temperature against AMO for each GCM (circles) for PI and LMH experiments compared with reconstructions (triangles). The mean correlations for the 7 GCMs are marked with black circles connected by a solid line. AMOnr and AMOr are the two definitions of AMO index described in Fig 1, while 'vo' indicates that the volcanic influence on East Asian temperature has been removed by linear regression in the LMH experiments. The thick circles and triangles show values significant at 95% level using a two-tailed student t-test. The reconstruction correlations are between the WN18 E Asian temperature and the three AMO reconstructions (Table 2; the correlation with WN17O is weak (0.24) because it represents only internal AMO variability).**

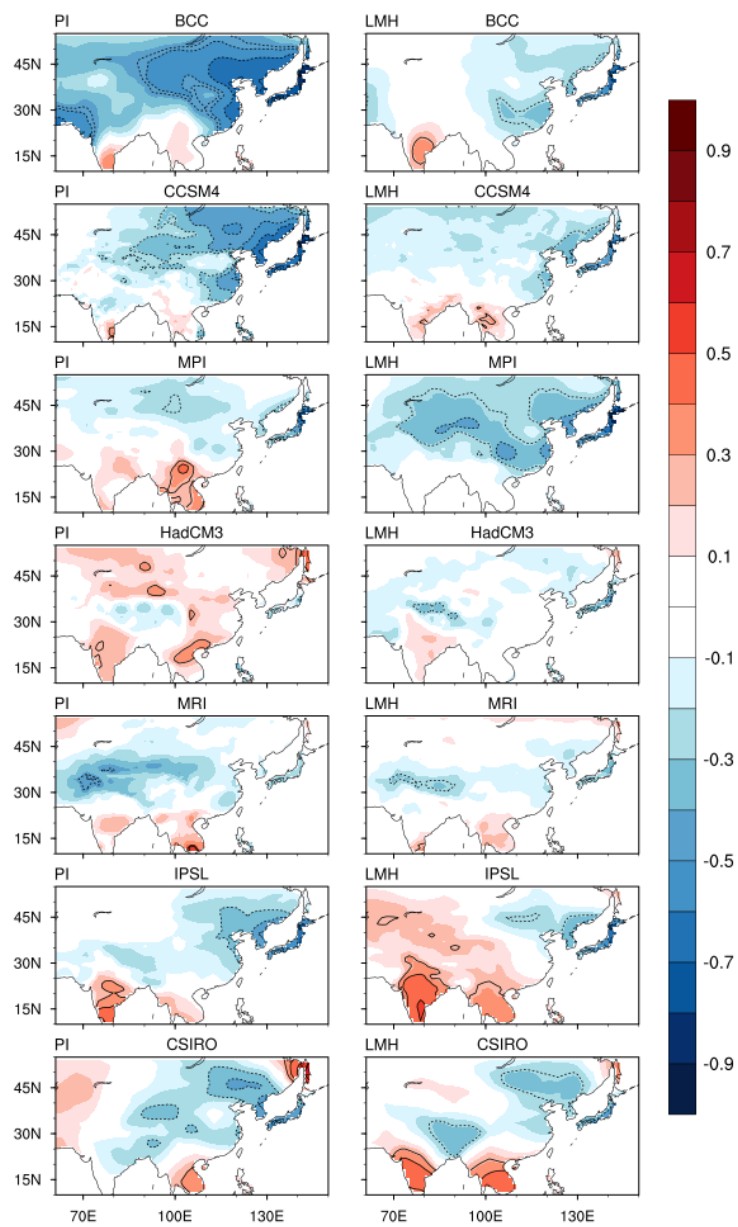

**Figure 5: Correlations between annual mean East Asian surface temperatures and PDO index for each GCM for PI (1st column) and LMH (2nd column) experiments. Correlation values significant at 90% levels using two-tailed Student's t-test are contoured.**

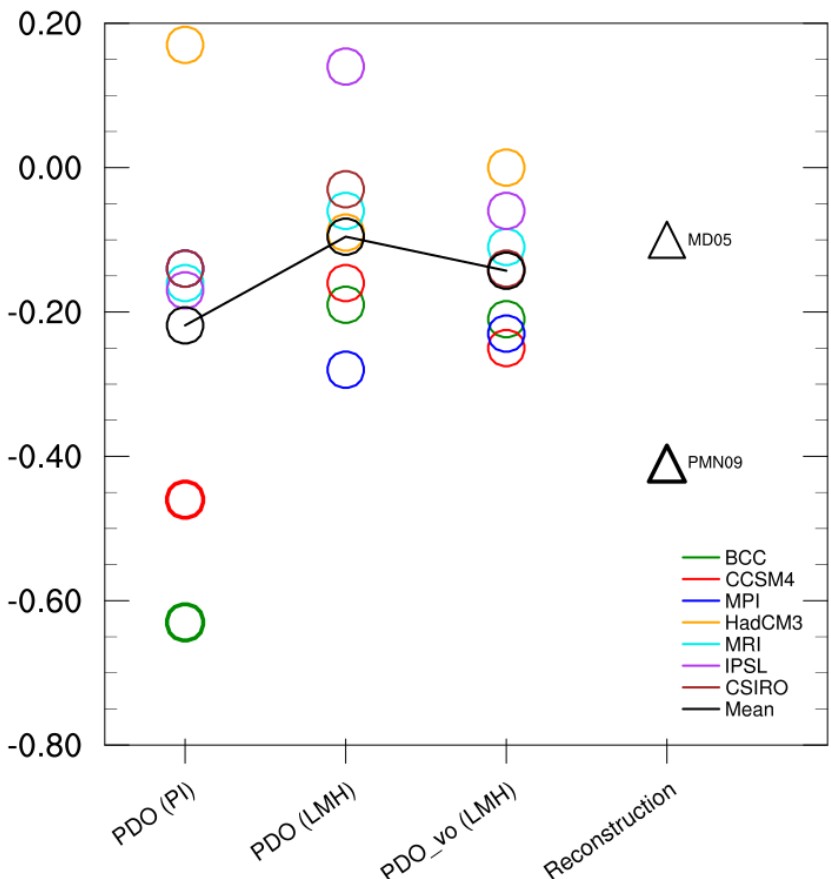

**Figure 6: Correlations of annual mean East Asian regional average surface temperature against PDO index for each GCM (circles) for PI and LMH experiments compared with reconstructions (triangles). The mean correlations for the 7 GCMs are marked with black circles connected by a solid line; 'vo' indicates that the volcanic influence on East Asian temperature has been removed by linear regression in the LMH experiments. The thick circles and triangles show values significant at 95% level using a two-tailed student t-test. The reconstruction correlations are between the WN18 E Asian temperature and the two PDO reconstructions (Table 2).**

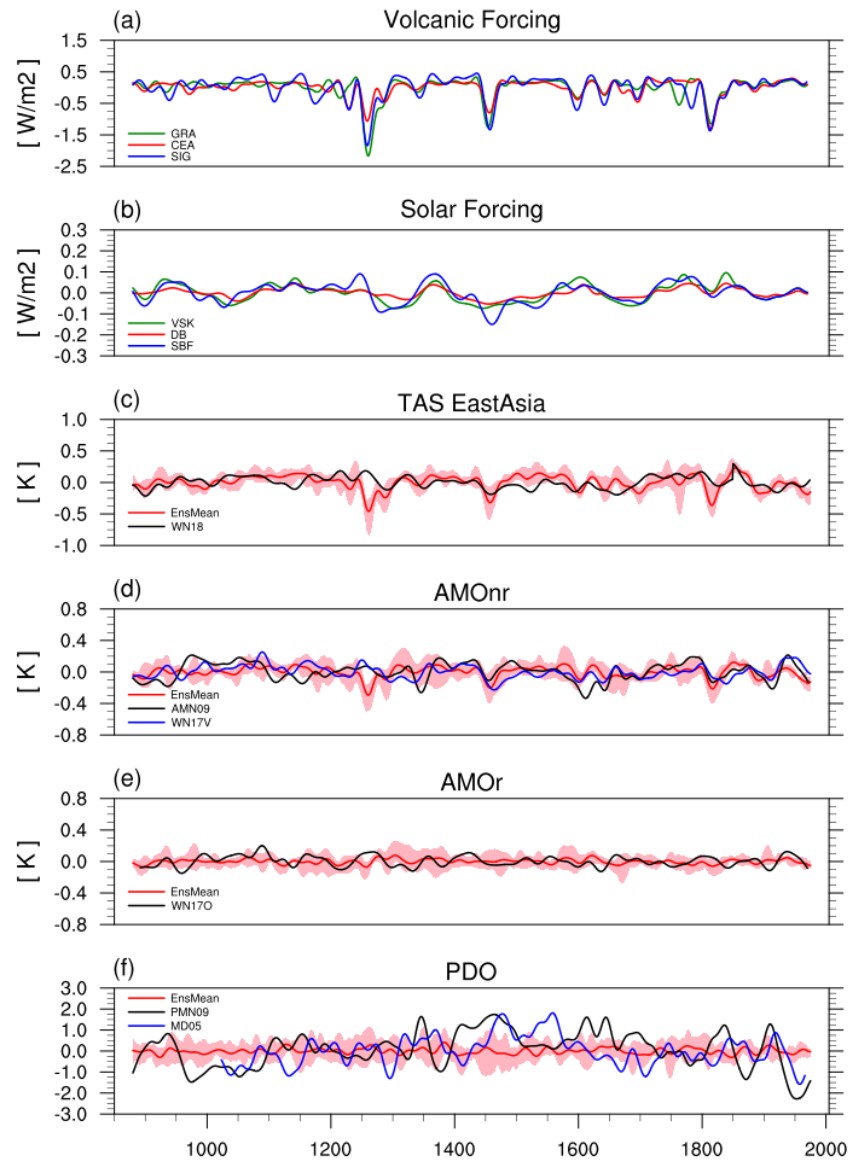

**Figure 7: Timeseries of (a) volcanic forcing, (b) solar forcing, (c) surface temperature over East Asia, (d) AMO index without subtracting the global SST (AMOnr), (e) AMO index after subtracting the global SST (AMOr), and (f) PDO index, all are annual mean values, passed through a 30-year low pass filter and truncated to remove filter end effects. Model simulations in (c)–(f) are given as the mean (red line) and spread (pink shading) of the 7 GCMs. Model simulations for surface temperature, AMO and PDO are compared with the available reconstructed data (black and blue lines).**

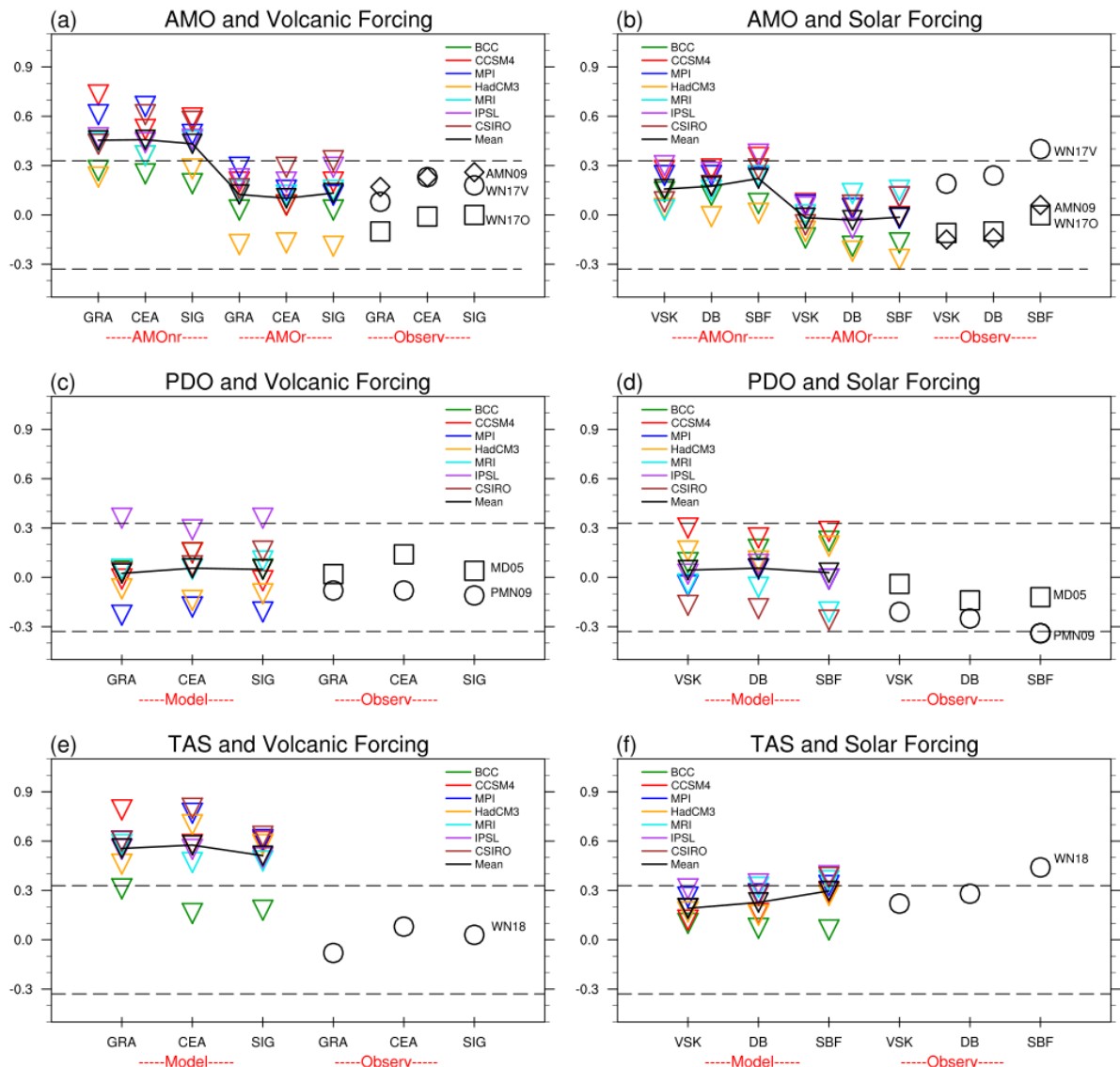

**Figure 8: The correlations of annual mean AMO, PDO and TAS with volcanic forcing (a, c, e) and solar forcing (b, d, f) for each GCM (triangles) and reconstruction (circle, square and diamond). The threshold values for the individual correlations significant at 95% level using a two-tailed student t-test are marked as dashed lines. The means of the 7 model correlations are shown by black triangles connected by black lines.**

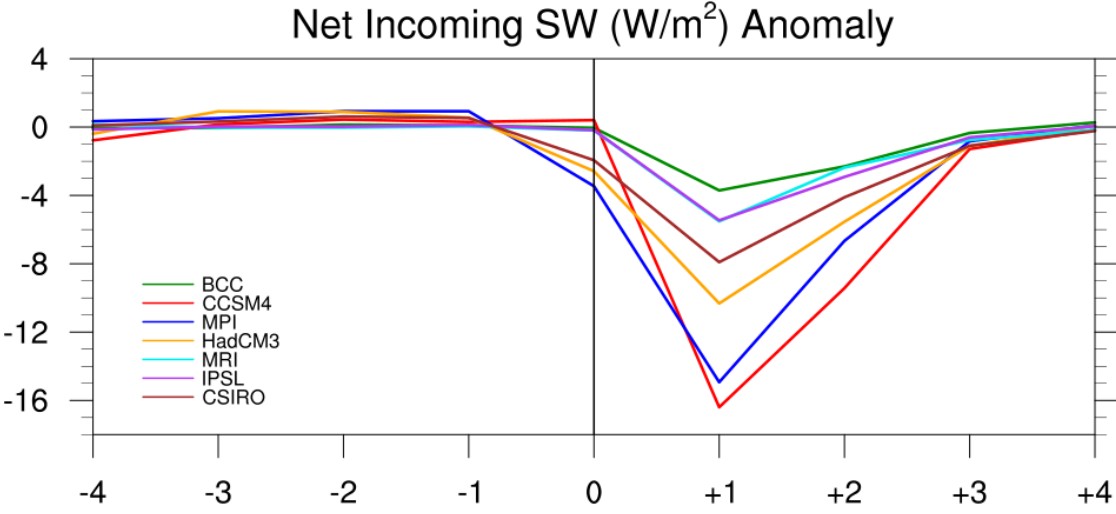

**Figure 9: Annual mean composite anomaly of net shortwave (SW) radiation at the top of the atmosphere for three large volcanic eruption events for each GCM. Individual GCM results are aligned so that their peak negative SW anomalies occur at year+1 (i.e. 1258, 1452, 1815 for BCC, CCSM4, IPSL; 1258, 1452, 1816 for MRI; 1258, 1456, 1816 for MPI, HadCM3, CSIRO).**

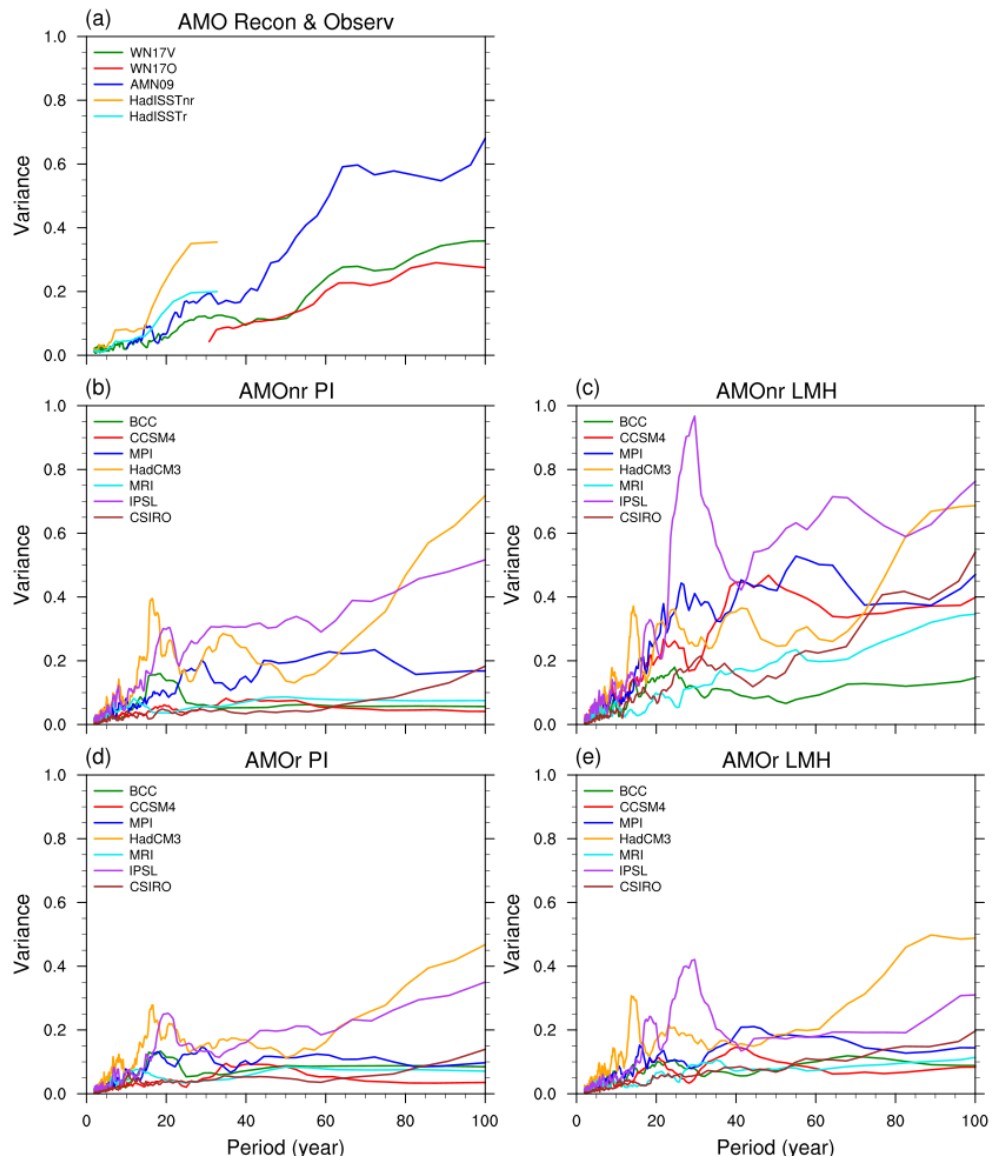

**Figure 10: Spectra of AMO index of reconstructions and instrumental data (a) and each of 7 GCMs for PI (b and d) and LMH (c and e) experiments. AMOnr and AMOr are the two definitions of AMO index described in Fig 1. Spectra are not shown for timescales that the underlying data cannot adequately represent (not for < 30 years for WN17O and not for < 10 years for AMN09 because these reconstructions use 30-year and 10-year low-pass filtered data respectively; not for > 30 years for HadISST because the instrumental data is too short to determine power on longer timescales).**

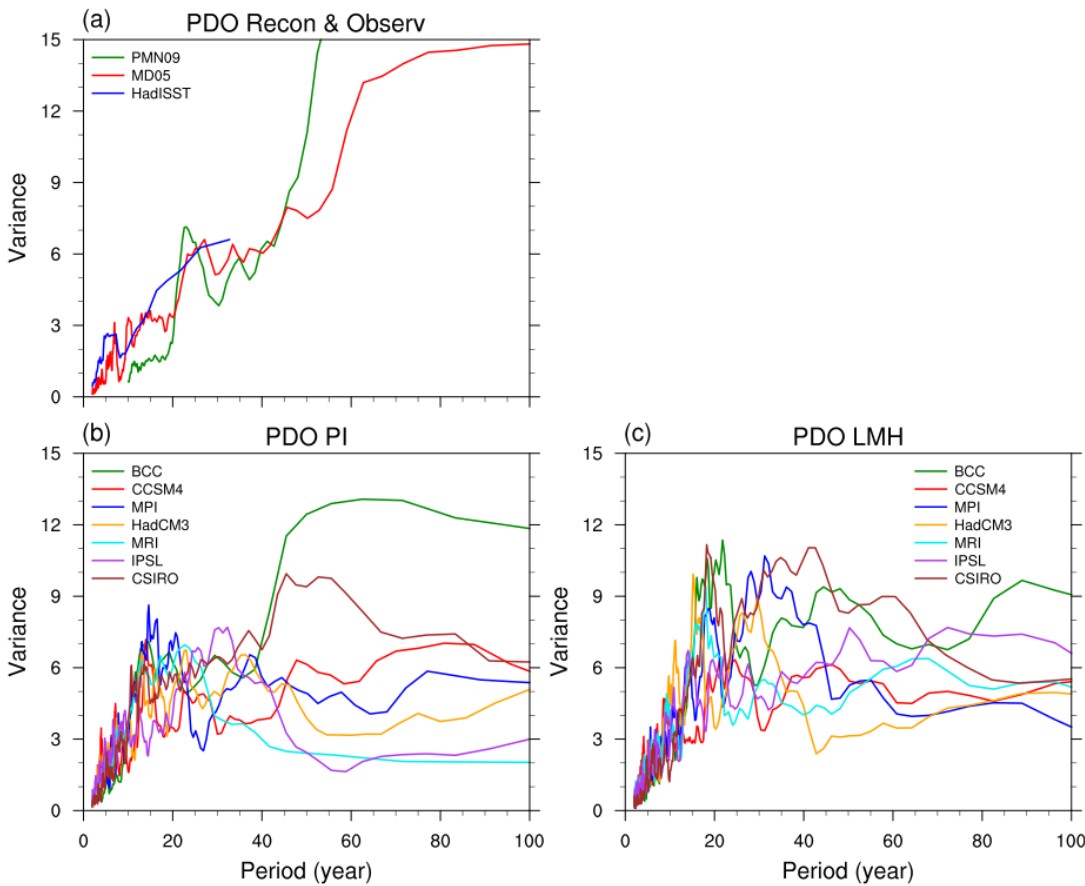

**Figure 11: Spectra of PDO index of reconstructions and instrumental data (a) and GCMs for PI (b) and LMH (c) experiments. Spectra are not shown for timescales that the underlying data cannot adequately represent (not for < 10 years for PMN09 because this reconstruction uses 10-year low-pass filtered data; not for > 30 years for HadISST because the instrumental data is too short to determine power on longer timescales).**