# Peer review of "Identifying teleconnections and multidecadal variability of East Asian surface temperature during the last millennium in CMIP5 simulations"

_Climate of the Past, 2018_

## Referee Comment (RC1) · Anonymous Referee #1 · 1 Jan 2019

Ratna et al., examined the relationships between AMO/PDO and surface temperature in East Asia (TAS) at multidecadal time scales based on models and reconstructions data, found that external forcing greatly strengthened the relationship between AMO and TAS but weakened relationship between PDO and TAS, and discussed the volcano influences. This is an interesting study on how external forcing influences on teleconnetions between AMO/PDO and TAS. However, I still have some concerns on this study.

Major concerns: 1) On the reliability of model and reconstruction data. Comparisons between modeled PDO/AMO from (CCSM4, MPI-ESM-P, BCC) with observed PDO/AMO index from HadISST/NCDC ERSST during the period of 1870-2000 should be added to evaluate the reliability of PDO/AMO index from model. There are several PDO/AMO reconstruction (such as, Gray et al., 2004; Shen et al., 2006). Although such PDO/AMO reconstructions are relatively short, the results seem more convincing by adding these records. In addition, there are published and robust Asian summer temperature reconstructions (e.g. Cook et al., 2013, Shi et al., 2015), such reconstruction data should be used. Comparisons among different reconstructions are as important as comparisons among the different models.

Shen, C., W.-C. Wang, W. Gong, and Z. Hao. 2006. A Pacific Decadal Oscillation record since 1470 AD reconstructed from proxy data of summer rainfall over eastern China. Geophysical Research Letters, vol. 33, L03702, 2006. Gray, S.T., L.J. Graumlich, J.L. Betancourt, and G.T. Pederson. 2004. A tree-ring based reconstruction of the Atlantic Multidecadal Oscillation since 1567 A.D. Geophysical Research Letters, 31:L12205, doi:10.1029/2004GL019932. Cook E R , Krusic P J , Kevin J. Anchukaitis et al. Tree-ring reconstructed summer temperature nomalies for temperate East Asia since 800 C.E. Climate Dynamics, 2013, 41(11-12):2957-2972. Shi F , Ge Q , Yang B , et al. A multi-proxy reconstruction of spatial and temporal variations in Asian summer temperatures over the last millennium. Climatic Change, 2015, 131(4):663-676.

2) On PDO signal. PDO has clear decadal and inter-decadal signal. Figure 11 also showed significant 15-20 years periods for PDO. However, all the time series are passed through a 30-year low pass filter using the Lanczos filter, which may miss key information of PDO. 10-year low pass filter should be used for PDO analysis.

3) On Volcano influences. Although previous studies showed that volcano eruptions affected decadal climate changes, it is equivocal that volcano eruptions affected multi-decadal climate changes. For example, TAS reconstruction showed clear volcanic forcing signal, and volcano eruptions resulting in pulses of cooler summer conditions that may persist for several years (See Figure 12 in Cook et al., 2013). However, this study

showed that there were not significant correlations between TAS and volcanic forcing (Figure 8c). In addition, superposed epoch analysis (SEA) should be used to test the impact of explosive volcanism on temperature.

4) On time scales of external forcing. there are other external forcings (e.g. solar activity) that should be considered. Solar activity has multi-decadal periods.

5) On influences of external forcing, external forcing greatly strengthened the relationship between AMO and TAS but weakened relationship between PDO and TAS. Do you think such results are related to definition and calculation of AMO and PDO? In simple terms, AMO reflects average SST, but PDO reflects spatial configuration of SST. So AMO may be related to external forcing while PDO may be related to internal variability.

Minor Concerns:

1) Page 3, Line 20-22. For temperature over East Asia, TAS reconstruction is summer temperature, and TAS model data is summer, cold season temperature and annual temperature. It is confusing. Please clarify which season temperature used in Figure 3-8 in Figure caption.

2) Page 13, Line 9, East Asiantemperature should be East Asian temperature.

3) Figure 7a, the label for y axis for volcanic forcing should be added.

4) Figure 8, confidence level explanation should be added.

---

## Author Comment (AC1) · 21 Jan 2019

Ratna et al., examined the relationships between AMO/PDO and surface temperature in East Asia (TAS) at multidecadal time scales based on models and reconstructions data, found that external forcing greatly strengthened the relationship between AMO and TAS but weakened relationship between PDO and TAS, and discussed the volcano influences. This is an interesting study on how external forcing influences on teleconnetions between AMO/PDO and TAS. However, I still have some concerns on this study.

Reply: We thank the referee for their time and for their helpful suggestions.

Major concerns:

1) On the reliability of model and reconstruction data. Comparisons between modeled PDO/AMO from (CCSM4, MPI-ESM-P, BCC) with observed PDO/AMO index from HadISST/NCDC ERSST during the period of 1870-2000 should be added to evaluate the reliability of PDO/AMO index from model. There are several PDO/AMO reconstruction (such as, Gray et al., 2004; Shen et al., 2006). Although such PDO/AMO reconstructions are relatively short, the results seem more convincing by adding these records. In addition, there are published and robust Asian summer temperature reconstructions (e.g. Cook et al., 2013, Shi et al., 2015), such reconstruction data should be used. Comparisons among different reconstructions are as important as comparisons among the different models.

Shen, C., W.-C. Wang, W. Gong, and Z. Hao. 2006. A Pacific Decadal Oscillation record since 1470 AD reconstructed from proxy data of summer rainfall over eastern China. Geophysical Research Letters, vol. 33, L03702, 2006. Gray, S.T., L.J. Graumlich, J.L. Betancourt, and G.T. Pederson. 2004. A tree-ring based reconstruction of the Atlantic Multidecadal Oscillation since 1567 A.D. Geophysical Research Letters, 31:L12205, doi:10.1029/2004GL019932. Cook E R , Krusic P J , Kevin J. Anchukaitis et al. Tree-ring reconstructed summer temperature nomalies for temperate East Asia since 800 C.E. Climate Dynamics, 2013, 41(11-12):2957-2972. Shi F , Ge Q , Yang B , et al. A multi-proxy reconstruction of spatial and temporal variations in Asian summer temperatures over the last millennium. Climatic Change, 2015, 131(4):663-676.

Reply: Evaluation of model PDO/AMO. We have made a spatial comparison of the models' PDO/AMO signatures with those observed PDO/AMO, using HadISST and ERSST. Their spatial patterns are in Figs. 1 and 2 of this reply (see below). We have used the HadISST data for the period 1871-2000 and ERSST for the period 1854-2000 based on the availability of the data. It can be seen that the modelled PDO/AMO

is agrees closely with observed PDO/AMO. We will use these extended figures (with the additional row showing the observations) in our revised manuscript.

Analysis of additional, shorter PDO/AMO reconstructions. The primary focus of this study is on the model simulations and the influence of external forcings, and on records of at least 1000-year length. While analysis of additional reconstructions is worth doing, that is more suited to another study examining shorter periods. We would prefer not to dilute our focus by analysis of additional reconstructions that are shorter (and, since our focus is also exclusively on multidecadal timescales and longer, having millennial-length timeseries is beneficial). Concerning the Gray et al. AMO reconstruction, we note the comments of Wang et al. (2017: disclosure, several authors are also authors of the current study): "The reconstruction of Gray et al. is based on a sparse tree-ring network, completely independent of our predictors; it has precise dating control, but its smaller network (only 12 sites) may compromise its representation of AMV if the centres of climate impact of AMV shift through time (also see the discussions in ref. 16)."

Wang J, Yang B, Ljungqvist FC, Luterbacher J, Osborn TJ, Briffa KR and Zorita E (2017) Internal and external forcing of multidecadal Atlantic climate variability over the past 1,200 years. Nature Geoscience 10, 512-517 (doi:10.1038/ngeo2962).

Analysis of individual Asian summer temperature reconstructions. The E Asian temperature reconstruction we used, from Wang et al. (2018), is a composite of seven published reconstructions that already includes the two suggested by the referee (Cook et al. 2013, Shi et al. 2015). The time series comparison of these three datasets can be seen in Wang et al (2018). Nevertheless, we have calculated the correlations between three of the individual summer temperature reconstructions (Cook et al. 2013, Shi et al. 2015, Zhang et al. 2018) and AMO (Wang et al. 2017, Mann et al. 2009), PDO (Mann et al. 2009, MacDonald et al. 2005), volcanic (GRA, Gao et al. 2008; CEA, Crowley et al. 2008; SIG, Sigl et al. 2015) and solar forcing (VSK, Vieira et al. 2011; SEA, Shapiro et al. 2011; SBF; Steinhilber et al. 2009) (Figs. 3 and 4 of this reply). They do show

some interesting differences, perhaps related to how well they resolve the response to volcanic forcing, so we will include these additional results in our revised manuscript. However, with the exception of the correlations between E Asian temperature and the Mann et al. (2009) AMO index, the only significant correlations (with some solar forcing, PDO and AMO reconstructions) at the multidecadal timescales are with the Wang et al. (2018) composite reconstruction.

Vieira, L. E. A., Solanki, S. K., Krivova, N. A., and Usoskin, I.: Evolution of the solar irradiance during the Holocene, Astron.Astroph., 531, A6, doi:10.1051/0004-6361/201015843, 2011. Shapiro, A. I., Schmutz, W., Rozanov, E., Schoell, M., Haberreiter, M., Shapiro, A. V., and Nyeki, S.: A new approach to the long-term reconstruction of the solar irradiance leads to large historical solar forcing, Astron. Astrophys., 529, A67, doi:10.1051/0004-6361/201016173, 2011. Steinhilber, F., Beer, J., and FrÂĺohlich, C.: Total solar irradiance during the Holocene, Geophys. Res. Lett., 36, L19704, doi:10.1029/2009GL040142, 2009.

2) On PDO signal. PDO has clear decadal and inter-decadal signal. Figure 11 also showed significant 15-20 years periods for PDO. However, all the time series are passed through a 30-year low pass filter using the Lanczos filter, which may miss key information of PDO. 10-year low pass filter should be used for PDO analysis.

Reply: We agree that PDO has a decadal signal which can been identified in Figure 11. However, the specific purpose of our study is to look at variability on multidecadal timescales (title and first line of the abstract), not decadal timescales, which is why we have passed all timeseries (including the PDO) through a 30-year low-pass filter. Therefore we do not use a 10-year filter because that would conflict with the aim of our study.

3) On Volcano influences. Although previous studies showed that volcano eruptions affected decadal climate changes, it is equivocal that volcano eruptions affected multidecadal climate changes. For example, TAS reconstruction showed clear volcanic

forcing signal, and volcano eruptions resulting in pulses of cooler summer conditions that may persist for several years (See Figure 12 in Cook et al., 2013). However, this study showed that there were not significant correlations between TAS and volcanic forcing (Figure 8c). In addition, superposed epoch analysis (SEA) should be used to test the impact of explosive volcanism on temperature.

Reply: Again, we note the aim of our study is explicitly to look at multidecadal timescales, so a superposed epoch analysis would not add more to the results already found. However, as noted above, we have now also analysed two individual E Asian temperature reconstructions, including Cook et al. (2013). Although the correlations with volcanic forcing are slightly stronger for Cook et al. (2013) than for the other reconstructions (Fig. 3 of this reply), they are still not statistically significant at the multidecadal timescale. This contrasts with two of the three climate models (CCSM4 and MPI), which show significant multidecadal correlations between simulated E Asian T and volcanic forcing, a finding that we report in the paper. There is also evidence that volcanic eruptions affect heat content and SST on these longer timescales (e.g. Gleckler et al. 2006). As suggested by the referee, the volcanic influence on the reconstructed temperatures is probably limited to the interannual timescale – but this timescale is not the focus of our study.

Gleckler, P. J., T. M. L. Wigley, B. D. Santer, J. M. Gregory, K. AchutaRao, and K. E. Taylor, 2006: Volcanoes and climate: Krakatoa's signature persists in the ocean. Nature, 439, 675.

4) On time scales of external forcing. there are other external forcings (e.g. solar activity) that should be considered. Solar activity has multi-decadal periods.

Reply: We focussed on volcanic forcing because it had previously been established that this was the largest external influence on the last millennium simulations (see the first sentence of section 5 of our manuscript). However, since we are also looking at reconstructions, the referee is correct that we should not neglect solar forcing, because

the reconstructions might show a significant association with solar forcing (indeed, as Wang et al. 2018 showed) even though the models may not. We have now included solar forcings in our analysis of the correlations between E Asian temperature and PDO/AMO/external forcing, both in model and reconstructions (right-hand column of Fig. 4 of this reply). We used three different solar forcing reconstructions (Vieira et al. 2011; Shapiro et al. 2011; Steinhilber et al. 2009). The only E Asian temperature reconstruction with significant multidecadal correlations to solar forcing is the composite reconstruction of Wang et al. (2018). We will include these additional results in our revised manuscript.

5) On influences of external forcing, external forcing greatly strengthened the relationship between AMO and TAS but weakened relationship between PDO and TAS. Do you think such results are related to definition and calculation of AMO and PDO? In simple terms, AMO reflects average SST, but PDO reflects spatial configuration of SST. So AMO may be related to external forcing while PDO may be related to internal variability.

Reply: Yes, the referee's explanation is correct. Minor Concerns: 1) Page 3, Line 20-22. For temperature over East Asia, TAS reconstruction is summer temperature, and TAS model data is summer, cold season temperature and annual temperature. It is confusing. Please clarify which season temperature used in Figure 3-8 in Figure caption.

Reply: The annual mean temperature is used in Figures 3-8. We will amend the figure captions to make this clear. 2) Page 13, Line 9, East Asiantemperature should be East Asian temperature.

Reply: We will correct this typo. 3) Figure 7a, the label for y axis for volcanic forcing should be added.

Reply: We will add the y-axis label for volcanic forcing: 'Radiative forcing (W m-2)' 4) Figure 8, confidence level explanation should be added.

[Figure]

Reply: We will add 'The bars marked with 'star' marks are significance at 95% using a two-tailed student t-test'.

Please also note the supplement to this comment:
https://www.clim-past-discuss.net/cp-2018-164/cp-2018-164-AC1-supplement.pdf

[Figure]

**Figure 1: The AMO spatial patterns.** Regression of annual-mean SST (K) on the AMO index for the three CMIP models for the LMH and PC experiments, compared with HadISST (d, h) and ERSST observation (i, p). (a-d) and (i-l) use the AMO index calculated without subtracting the global SST anomaly while (e-h) and (m-p) use the AMO index after subtracting the global SST anomaly.

**Fig. 1.**

[Figure]

Figure 2: The PDO spatial pattern. The leading EOF of North Pacific SST anomalies for the three CMIP models, PC (a-c) and LMH (d-f) simulations, compared with HadISST and ERSST observation.

**Fig. 2.**

[Figure]

**Figure 3: Correlation of TAS reconstructions with AMO, PDO, volcanic forcing and solar forcing reconstructions. The bars marked with 'star' are significance at 95% level using a two-tailed student t-test.**

**Fig. 3.**

[Figure]

Figure 4: The correlations of (a) AMO, (b) PDO and (c) TAS with volcanic and solar forcing (models and reconstructions) data. (a-c) for volcanic forcing and (d-f) for solar forcing. The bars marked with 'star' are significance at 95% level using a two-tailed student t-test.

**Fig. 4.**

---

## Referee Comment (RC2) · Anonymous Referee #2 · 9 Apr 2019

General Comments: Ratna et al. examine the influence of transient external forcing (volcanic eruptions) on PDO and AMO variability and teleconnection patterns as they relate to East Asian surface air temperatures (SAT) in three PMIP3/CMIP5 past1000 simulations and paleoclimate reconstructions. This is an interesting study, and the results have interesting implications for how external forcing can impact internal variability and teleconnections. However, more work is needed to compare model output to observations and expand the study to other models.

Main Concerns:

[Figure]

1) There are at least ten CMIP5/PMIP3 past1000 (Last Millennium) simulations available on ESGF that span the 850-1849 CE time period (BCC, CCSM4, CSIRO, FGOALS, GISS, Had, IPSL, MIROC, MPI, MRI). The authors exclude several of these simulations (MIROC, FGOALS, GISS) due to spin up/model drift/trend issues and cite Atwood et al for why they exclude these simulations. However, the authors choose not to use the output from CSIRO, HadCM3, IPSL, or MRI (some of which are included in the analysis of Atwood et al). The results therefore seem incomplete and selectively presented- why the exclusion of these other simulations? Please include analyses of these other Last Millennium simulations or at least provide a reason for why these other Last Millennium simulations have been excluded (the data have been available for at least 8-12 months online, so I hope it's not a data availability issue?). As the manuscript is currently written, 1/3 of the models show a completely different result, but this is only one model- is this really 1/3 of all CMIP5 Last Millennium models, or just one outlier in the CMIP5 Last Millennium simulations?

2) The authors concatenate the Last Millennium (∼850-1849CE) and the Historical simulations (∼1850-2005CE) after removing the linear trend from each of these time segments separately. Removing a linear trend from either instrumental or CMIP5 data over the entire 1850-2005CE time period can be problematic if the main component of the 'warming trend' is in the 20th century. Multiple papers choose to remove the linear trend over the 20th century only (e.g., Deser et al., 2010; Messie and Chavez, 2011; Franzke, 2014, Nature Climate Change; Ji et al., Nature Climate Change, 2014). Similarly, many CMIP5 historical simulations appear to show much of the global warming trend starting in the 20th century, so removing a trend over the full historical simulation period (1850-2005) may add in decadal-centennial variability. To avoid this detrending and concatenation problem, could the analysis just be conducted over the 850-1849CE time period (especially because it seems the authors are mostly focused on the impacts of volcanic eruptions on the PDO and AMO in the pre-1850CE time period?). Some recent work even suggests that the dynamics of the system change once GHG forcing becomes dominant (e.g., Song and Yu, 2015, J Clim; Brown et al., 2017, Nature

Climate Change), so including this time period could be arguably problematic.

3) There is no comparison between the spatial patterns of the AMO and PDO in instrumental-based reconstructions and the three models used here- perhaps some of these simulated spatial patterns are more realistic than others? The authors state that the model results are realistic, but never show this in the manuscript. The Climate Variability Diagnostics Package (http://webext.cgd.ucar.edu/Multi-Case/CVDP_ex/CMIP5-Historical/) shows that the spatial expressions of the AMO (and PDO) can be quite different in the various CMIP5 Historical simulations. Interpretation of the model results may be viewed through a more informed perspective if the models are compared to instrumental-based observations.

4) Varying significance levels are used in the paper (90% vs 95%). Please use a consistent 95% or 99% significance level- as the paper stands, it appears that the significance level has been lowered to show 'significance spectral peaks' (e.g., Fig 10), but the spectra barely surpass this 90% level- why not use 95% or 99% everywhere? At least please include some discussion of the sensitivity of the results to significance level if the results don't pass this higher threshold (significance levels are admittedly is arbitrary, but the current, inconsistent use of 90% runs the risk of appearing selectively low to attempt to present a 'significant' result).

General minor issues: Many authors abbreviate pre-industrial Control as PI (e.g., Atwood et al., 2016, J Clim, among others)- in an effort to maintain some sort of standard abbreviation that may be quickly recognized, I would encourage the authors to employ more commonly used acronyms (e.g., PI or piControl).

Also, when reading through the figures, it is difficult to interpret the acronyms used in each figure without searching through the other figure captions or the text for the definitions of the acronyms- please define the acronyms used in each figure in each figure caption (or at least reference where they are defined) so readers can quickly understand the figure without searching for what they mean.

Specific comments: Page 1, Line 12-13: The simulated PDO and AMO spectra and spatial patterns are never compared to instrumental-based patterns or spectra (or even to proxy-based spatial patterns). Please include figures/analysis that support this statement in the main text or remove it.

Page 2, ~line 10: The previous paragraph critiques the instrumental and proxy-based records, but little attention is paid to potential model deficiencies- can you at least briefly discuss or cite a few papers that may critique or even acknowledge that CMIP5/PMIP3 models have their own biases and problems as they relate to low-frequency SAT variability (e.g., Laepple and Huybers, 2014; Parsons et al., 2017 J Clim; ) or 'modes' of internal variability (or their responses to stratospheric aerosol loading from volcanic eruptions)? Alternatively, directing the reader to where these model deficiencies, and their implications for your results, are going to be discussed later in the paper could be helpful.

Page 3, lines 5-6: please see general comments in previous section- why were the bulk of the CMIP5/PMIP3 Last Millennium simulations excluded? Analysis of results would appear much more robust if an attempt is made to present more than 1/3 of the Last Millennium simulations, or if reasoning can be given why the other simulations were excluded. Also, what is the cutoff used for a drift that is 'too strong'? Is this a global or local drift? All the CMIP5/PMIP3 past1000 simulations appear to show some sort of trend/drift at many grid points- the question is what is too much for the purposes of this AMO/PDO teleconnection study. Please clarify.

Page 3, Line 20: please see general comments in previous section- removing one linear trend over the full 1850-2005CE time period seems like it may add in low-frequency variability, and I am still not even sure why the historical simulations have been included if the focus is on the impact of volcanic eruptions in the pre-historical simulation time period.

Page 4, Line 8: 'we don't see much differences'- this is a subjective statement. What

criteria are used? Perhaps something like a pattern metric or Euclidean distances metric could be used to say something more quantitative?

Page 4, Line 10: Please explain how the TAS time series is made- I assume annual mean (Jan-Dec?) temperature at each grid box, latitude-weighted, and masked ocean grid boxes? Over what latitude and longitude range is this area average made (is it the whole region used in the maps in the figures showing East Asia?)? Please provide more details in the text.

Page 5, line 5-7: There are other PDO reconstructions- fine to not include them, but can you state why this one is selected over others?

Page 5, lines 11-15: As far as I can tell, the model-based PDO indices are made from monthly data, and the paleo-based PDO indices are made 'annual' data (or seasonally sensitive proxy records)- would a better comparison be to make annual means of SAT for the model data, then construct the PDO index, so the index is more comparable to the annual proxy-based index? (or can you show that the annual and monthly model-based PDO patterns and time series are similar?)

Page 5, Line 21, line 25: The 90% significance level seems oddly low, and arbitrarily used in only certain cases- do your results consistently pass a 95% significance test (both the regions in the maps and the spectra)? For example, the 'significant' spectral peaks in Figure 10 appear quite close to the 90% significance level- if you made this a 95 or 99%, are these 'significant spectral peaks' at all significant?

Page 8, line 26: the authors discuss a weak response in BCC to volcanic eruptions- is this a finding that has been noted previously (e.g., Driscoll et al., JGR, 2012, or some sort of similar CMIP5 comparison to observations?)? How realistic is this model's response relative to the other models' responses to volcanic eruptions (especially compared to observations of more recent eruptions and their impacts)? I ask because this difference seems to be important to the results- for example, should the BCC changes (or lack thereof relative to the other models) in PDO, AMO, and associated teleconnections with E Asia be viewed as just as realistic as the other models' responses? Or is it an outlier because it doesn't respond at all to volcanic eruptions when it should?

Page 9, Line 25-26: It would be helpful to show results from the other four CMIP5 Last Millennium simulations here to put these results in context- right now, 1/3 of the models show a completely different result, but this '1/3 of models' is just the BCC model.

Page 9, Line 26-29: Would this result imply that the models show an unrealistically large response to eruptions? Or that there is too little internal, low-frequency variability (e.g., Laepple and Huybers)? Or does this suggest both, or something else?

Page 10, line 2-3: the authors state that 'all models display red spectra'- in the methods (and in the time series in the figures), it seems that the data have been low-pass filtered, so by definition, the high-frequency variability has been reduced relative to the low-frequency variability (thus reddened)- I'm not sure that 'redness' really means anything in this case. If 'redness' does mean something after the data have been filtered, or if the data have not been low-pass filtered before spectral estimation, please clarify/explain- for example, if the authors mean to say that one model has more low-frequency variability than another, that may be more accurate.

Furthermore, the 'pronounced multidecadal variability' barely surpasses the 90% significance threshold, as do most of the 'significant' peaks referenced in this paragraph.

These power spectra (AMO and PDO power spectra figures) are shown without any error bars- when the spectra are compared and declared similar/different, some sort of spectral estimation confidence bound/error bar on the figure could show if these differences fall within the confidence bounds of the spectral estimates.

Page 10, Lines ∼5-15: Perhaps this is the first time that this analysis has been done, but I would be surprised- has anyone else compared the power spectra across these simulations before? For example, Cheung et al., (2017) compares instrumental-based AMO and Pacific variability to CMIP5 historical simulations (and also how the spatial

patterns associated with these modes can change through time). Parsons et al., 2017 (J Clim) compares instrumental, AR1, and CMIP5 Last Millennium, and CMIP5 Control spectra over the North Pacific and North Atlantic, and Fredriksen and Rypdal (2016, J Clim) compare spectra over ocean basins in CMIP5 models.

Page 10, Line 23: the authors claim that the spatial patterns of AMO and PDO are similar to the patterns from observations. I see no comparisons among modeled and observed spatial patterns of variability. In fact, it would be helpful if the authors would show the spatial patterns from observations (of course acknowledging that the instrumental-based data have their own limitations) in Figures 1 and 2- this would help put the model results in context.

Page 10, line 25: again, it's unclear if the data have been low-pass filtered before spectral analysis. Also, see my above comments- saying the spectra are 'red' seems meaningless if the data have been low-pass filtered. Again, the significant peaks barely surpass a 90% threshold- please discuss or mention if this significance is sensitive to threshold level.

Also, as stated above it would be good to include error bars/lines on the spectra to know if the 'significant' differences from the background spectrum significant given uncertainties in the power spectral estimation?

Page 11, ~Line 25: good point.

Page 12, ~line 4: OK, so other recent methods have been used to reconstruct SAT fields (e.g., Last Millennium Reanalysis from Hakim et al., 2016, JGRA and Tardiff et al., in review at CP)

Figures: Figures 1, 2, 3, 5: please include panels showing similar analyses from instrumental-based data products.

Figure 4, Figure 6: it is interesting to see the PDO-E Asia and AMO-E Asia differences, but it would be nice to see some confidence bars on the control run values. For example, Coats et al. 2013 show that teleconnections can change from century to century. Could you do some sort of running correlation or subsample the control run to see how variable this E Asian relationship is (or is there enough data?)

Figure 9: Is there a way to put these results in context? For example, if you include the post-Pinatubo response in these models, could you show how the models compare to obs? Which models are more realistic? (CCSM4/MPI or BCC?)

Figures 10 and 11: inclusion of instrumental-based spectra could be helpful here too-how realistic are these reconstructions?

Compact listing of purely technical corrections (typing errors, etc.). Page 1, Line 13: 'and their spectral characteristics'- remove 'their' Page 3, line17: change sentence to: 'Each model version was the same across all the simulations.' Page 4, line 8: 'much differences'- please re-word (e.g., 'A pattern correlation statistic shows minimal differences among …' Page 5, Line 7: 'largely suffer from the influence of external forcing' Page 6, Line 4: 'no time-varying (transient?) external radiative forcing' Page 6, line 31: 'This situation is equivalent to (that?) of Fig.' – there appears to be a missing word here Page 7, Line 13-15: 'in the southern parts'…'in all three models'…'with the strongest correlation in the northeast region' Page 7, line17: 'though it varies' Page 9, Line 10: the sentence starting with 'Despite' appears a bit awkward- suggest rewording

---

## Author Comment (AC2) · 16 May 2019

General Comments: Ratna et al. examine the influence of transient external forcing (volcanic eruptions) on PDO and AMO variability and teleconnection patterns as they

relate to East Asian surface air temperatures (SAT) in three PMIP3/CMIP5 past1000 simulations and paleoclimate reconstructions. This is an interesting study, and the results have interesting implications for how external forcing can impact internal variability and teleconnections. However, more work is needed to compare model output to observations and expand the study to other models. Reply: We are grateful to the referee for their careful review and that they consider our work to be of interest. We respond below to the suggestions for expanding the scope of the work.

Main Concerns: 1) There are at least ten CMIP5/PMIP3 past1000 (Last Millennium) simulations available on ESGF that span the 850-1849 CE time period (BCC, CCSM4, CSIRO, FGOALS, GISS, Had, IPSL, MIROC, MPI, MRI). The authors exclude several of these simulations (MIROC, FGOALS, GISS) due to spin up/model drift/trend issues and cite Atwood et al for why they exclude these simulations. However, the authors choose not to use the output from CSIRO, HadCM3, IPSL, or MRI (some of which are included in the analysis of Atwood et al). The results therefore seem incomplete and selectively presented- why the exclusion of these other simulations? Please include analyses of these other Last Millennium simulations or at least provide a reason for why these other Last Millennium simulations have been excluded (the data have been available for at least 8-12 months online, so I hope it's not a data availability issue?). As the manuscript is currently written, 1/3 of the models show a completely different result, but this is only one model- is this really 1/3 of all CMIP5 Last Millennium models, or just one outlier in the CMIP5 Last Millennium simulations?

Reply: We originally considered al six models that had CMIP5/PMIP3 Last Millennium and CMIP5 historical simulations and that had data for all the necessary variables in the UK JASMIN facilities. We then discarded three due to drift issues as explained in our manuscript. Following the referee's recommendation, we sought data for the additional models suggested from other parts of the ESGF and we have obtained sufficient data to extend our study to another three CMIP5 models (HadCM3, MRI and IPSL) and we are downloading the necessary data to include CSIRO-Mk3L-1-2 as well. We have already

applied many parts of our analysis to these additional models and have included the results in updated figures in this reply (Figs. R1-R4). We agree that the manuscript would be strengthened by revising it to include these additional model results. Some of the results are similar, but there are some differences in the correlations with E Asian temperature that we would discuss in a revised manuscript and we are happy to make these changes.

2) The authors concatenate the Last Millennium (_850-1849CE) and the Historical simulations (_1850-2005CE) after removing the linear trend from each of these time segments separately. Removing a linear trend from either instrumental or CMIP5 data over the entire 1850-2005CE time period can be problematic if the main component of the 'warming trend' is in the 20th century. Multiple papers choose to remove the linear trend over the 20th century only (e.g., Deser et al., 2010; Messie and Chavez, 2011; Franzke, 2014, Nature Climate Change; Ji et al., Nature Climate Change, 2014). Similarly, many CMIP5 historical simulations appear to show much of the global warming trend starting in the 20th century, so removing a trend over the full historical simulation period (1850-2005) may add in decadal-centennial variability. To avoid this detrending and concatenation problem, could the analysis just be conducted over the 850-1849CE time period (especially because it seems the authors are mostly focused on the impacts of volcanic eruptions on the PDO and AMO in the pre-1850CE time period?). Some recent work even suggests that the dynamics of the system change once GHG forcing becomes dominant (e.g., Song and Yu, 2015, J Clim; Brown et al., 2017, Nature Climate Change), so including this time period could be arguably problematic.

Reply: We recognise the referee's concerns about linearly detrending the Historical simulations, but our findings are not sensitive to this choice.

We note that we are trying to replicate in the models some aspects of what other studies have done using observations (proxy-based reconstructions and/or instrumental) and (a) in some cases linear detrending over the instrumental era is done even though it may not be optimal for the reasons given by the referee; and (b) the timing of the

start of the anthropogenic warming can be in conflict with a linear detrending that begins 1850 but this would be ameliorated for the PDO and for the AMOrem indices by the prior removal of global-mean SST from the Atlantic or Pacific SST values.

Nevertheless, we have tested to see whether our findings are sensitive to this issue by restricting the analysis to only the Last Millennium simulation and found that our results are quite similar to those we obtained by the combined detrended LM plus detrended historical simulations (compare columns of Fig. R5 and R6 for correlations with E Asia temperature for AMO and PDO, respectively).

We would report this sensitivity test in the revised manuscript and discuss the few small differences that do occur. We suggest to keep the main results based on the combined LM+Historical analysis because the benefits of having a longer series to analyse outweighs the concerns raised now that we have shown that our findings are not sensitive to this issue.

3) There is no comparison between the spatial patterns of the AMO and PDO in instrumental-based reconstructions and the three models used here- perhaps some of these simulated spatial patterns are more realistic than others? The authors state that the model results are realistic, but never show this in the manuscript. The Climate Variability Diagnostics Package (http://webext.cgd.ucar.edu/Multi-Case/CVDP_ex/CMIP5-Historical/) shows that the spatial expressions of the AMO (and PDO) can be quite different in the various CMIP5 Historical simulations. Interpretation of the model results may be viewed through a more informed perspective if the models are compared to instrumental-based observations.

Reply: We will include a comparison of the AMO and PDO patterns with the instrumental data (similar comment from referee 1), see Fig. R1 and R2 above.

4) Varying significance levels are used in the paper (90% vs 95%). Please use a consistent 95% or 99% significance level- as the paper stands, it appears that the significance level has been lowered to show 'significance spectral peaks' (e.g., Fig 10),

but the spectra barely surpass this 90% level- why not use 95% or 99% everywhere? At least please include some discussion of the sensitivity of the results to significance level if the results don't pass this higher threshold (significance levels are admittedly is arbitrary, but the current, inconsistent use of 90% runs the risk of appearing selectively low to attempt to present a 'significant' result).

Reply: Significance tests are reported for three types of analysis in the manuscript. (1) For correlations between area-averaged temperature and driving factors (bar charts) we used the 'standard' 95% level. (2) For correlations between temperature fields and driving factors (contoured maps) we lowered this to the 90% level because the additional noise at the grid cell level increases the risk of a type II error (wrongly failing to reject the null hypothesis that there is no correlation). (3) For power spectra of AMO and PDO, we used a 90% level, but actually our interest is not really in the significance of the individual spectral peaks (and whether they pass an arbitrary level or not) but in the overall shape of the spectra, their redness and broad multi-decadal power, and whether these are similar between models, with/without forcing, and between AMO index definitions. We will explain this better in a revised manuscript, we will also either remove the significance lines from the spectra or change them to 95%, and justify the use of 90% for the fields as explained above.

General minor issues: Many authors abbreviate pre-industrial Control as PI (e.g., Atwood et al., 2016, J Clim, among others)- in an effort to maintain some sort of standard abbreviation that may be quickly recognized, I would encourage the authors to employ more commonly used acronyms (e.g., PI or piControl).

Reply: We will modify the manuscript to use the abbreviation PI for pre-industrial Control.

Also, when reading through the figures, it is difficult to interpret the acronyms used in each figure without searching through the other figure captions or the text for the definitions of the acronyms- please define the acronyms used in each figure in each

figure caption (or at least reference where they are defined) so readers can quickly understand the figure without searching for what they mean.

Reply: Now we will define the acronym for each Figure captions.

Specific comments: Page 1, Line 12-13: The simulated PDO and AMO spectra and spatial patterns are never compared to instrumental-based patterns or spectra (or even to proxy-based spatial patterns). Please include figures/analysis that support this statement in the main text or remove it.

Reply: We will include a comparison of the AMO and PDO patterns with the instrumental data , see Fig. R1 and R2 above.

Page 2, _line 10: The previous paragraph critiques the instrumental and proxy-based records, but little attention is paid to potential model deficiencies- can you at least briefly discuss or cite a few papers that may critique or even acknowledge that CMIP5/PMIP3 models have their own biases and problems as they relate to low-frequency SAT variability (e.g., Laepple and Huybers, 2014; Parsons et al., 2017 J Clim; ) or 'modes' of internal variability (or their responses to stratospheric aerosol loading from volcanic eruptions)? Alternatively, directing the reader to where these model deficiencies, and their implications for your results, are going to be discussed later in the paper could be helpful.

Reply: we will cite these references and add a few sentences to describe their implications for potential model deficiencies. We will discuss that Laepple and Huybers (2014) found potential deficiencies in CMIP5 SST variability, with model simulations diverging from a multiproxy estimate of SST variability (that is consistent between proxy types and with instrumental estimates) toward longer timescales. Parsons et al. (2017) found very different pictures of natural variability between CMIP5 models, including the North Atlantic, and between models and paleoclimate data in the tropics, in terms of the magnitude and spatial consistency of climate variance across interannual to centennial timescales.

Laepple, T., and P. H. Huybers, 2014: Ocean surface temperature variability: Large model–data differences at decadal and longer periods. Proc. Natl. Acad. Sci. USA, 111, 16 682–16 687, https://doi.org/10.1073/pnas.1412077111.

Page 3, lines 5-6: please see general comments in previous section- why were the bulk of the CMIP5/PMIP3 Last Millennium simulations excluded? Analysis of results would appear much more robust if an attempt is made to present more than 1/3 of the Last Millennium simulations, or if reasoning can be given why the other simulations were excluded. Also, what is the cutoff used for a drift that is 'too strong'? Is this a global or local drift? All the CMIP5/PMIP3 past1000 simulations appear to show some sort of trend/drift at many grid points- the question is what is too much for the purposes of this AMO/PDO teleconnection study. Please clarify.

Reply: This comment has been addressed under the first "Main Concerns" earlier: we will extend our analysis to include four more models CSIRO, HadCM3, MRI and IPSL and we will revise the manuscript to include these models and compare the additional results.

Our decision to exclude three CMIP5 models (MIROC-ESM, FGOALS-s2 and GISS) was based on the results discussed in Atwood et al. 2016, Fleming and Anchukaitis, 2016; Bothe et al., 2013. This is mentioned in our original manuscript (Page 3, Line 7). Atwood et al (2016) and Bothe et al (2013) discussed long term drift in global mean surface air temperature in their PI simulations. Similarly, Fleming and Anchukaitis (2016) found drift in the Last Millennium simulations, which are apparent in the initial several centuries and excluded from their PDO analysis.

Page 3, Line 20: please see general comments in previous section- removing one linear trend over the full 1850-2005CE time period seems like it may add in low-frequency variability, and I am still not even sure why the historical simulations have been included if the focus is on the impact of volcanic eruptions in the pre-historical simulation time period.

[Figure]

Reply: This comment has been addressed under the second "Main Concerns" earlier. Our findings are not sensitive to this choice and we will include this in a revised manuscript. Our focus is broader than just the impact of volcanic eruptions, we are interested in the influence of external forcings in general on the diagnosis of the role of internal variability from observational evidence. During these simulations, volcanic forcing plays a major role so we did some additional analysis of that but it is not our only result.

Page 4, Line 8: 'we don't see much differences'- this is a subjective statement. What criteria are used? Perhaps something like a pattern metric or Euclidean distances metric could be used to say something more quantitative?

Reply: we will revise this sentence to be less subjective, noting which key features (position and strength of the loading maxima and loading gradients) of the PDO patterns are present in the simulated and observed fields.

Page 4, Line 10: Please explain how the TAS time series is made- I assume annual mean (Jan-Dec?) temperature at each grid box, latitude-weighted, and masked ocean grid boxes? Over what latitude and longitude range is this area average made (is it the whole region used in the maps in the figures showing East Asia?)? Please provide more details in the text.

Reply: For the TAS time series, the annual mean (Jan-Dec) TAS is calculated over the land grid points only and area averaged over the region 60E -150E and 10N-55N. We will add this additional information to a revised manuscript.

Page 5, line 5-7: There are other PDO reconstructions- fine to not include them, but can you state why this one is selected over others?

Reply: We have used the selected PDO reconstructions based on the availability of the data for a longer period that covers at least 1000 years of our main analysis period 850-2000.

[Figure]

Page 5, lines 11-15: As far as I can tell, the model-based PDO indices are made from monthly data, and the paleo-based PDO indices are made 'annual' data (or seasonally sensitive proxy records)- would a better comparison be to make annual means of SAT for the model data, then construct the PDO index, so the index is more comparable to the annual proxy-based index? (or can you show that the annual and monthly modelbased PDO patterns and time series are similar?)

Reply: The model simulated monthly mean SST data were, in fact, already converted to annual mean data before applying the EOF analysis to get the PDO pattern and its time series (i.e. as suggested by the reviewer – we will make this clear by adding 'annual-mean SST anomalies' to the Fig. 2 caption). We have done this because all our analysis is based on the annual mean data, which also compares with the annual mean reconstructed data. We have also confirmed that model based PDO patterns for annual and monthly data are similar (figure R7).

Page 5, Line 21, line 25: The 90% significance level seems oddly low, and arbitrarily used in only certain cases- do your results consistently pass a 95% significance test (both the regions in the maps and the spectra)? For example, the 'significant' spectral peaks in Figure 10 appear quite close to the 90% significance level- if you made this a 95 or 99%, are these 'significant spectral peaks' at all significant?

Reply: Please see our response to "main concern (4)" above.

Page 8, line 26: the authors discuss a weak response in BCC to volcanic eruptions- is this a finding that has been noted previously (e.g., Driscoll et al., JGR, 2012, or some sort of similar CMIP5 comparison to observations?)? How realistic is this model's response relative to the other models' responses to volcanic eruptions (especially compared to observations of more recent eruptions and their impacts)? I ask because this difference seems to be important to the results- for example, should the BCC changes (or lack thereof relative to the other models) in PDO, AMO, and associated teleconnections with E Asia be viewed as just as realistic as the other models' responses? Or is

it an outlier because it doesn't respond at all to volcanic eruptions when it should?

Reply: By analysing the CMIP5 historical simulations, Driscoll et al. (2012) found largest anomaly in the reflected SW radiation in the BCC model. Here, we show that the weak response in BCC to volcanic response only exists in the last millennium simulations, where we have analysed three major volcanic eruptions that happened in the last millennium. We also have analysed the same for the major volcanic events during the historical period but didn't find such weak response in BCC model compared to the other two models. So, we feel that the weak volcanic forcing and response only exist in BCC in their last millennium simulations.

Page 9, Line 25-26: It would be helpful to show results from the other four CMIP5 Last Millennium simulations here to put these results in context- right now, 1/3 of the models show a completely different result, but this '1/3 of models' is just the BCC model.

Reply: As noted earlier, we have now analysed this for three other models (HadCM3, MRI and IPSL), with a fourth underway (CSIRO) and would include these results in a revised manuscript. Page 9, Line 26-29: Would this result imply that the models show an unrealistically large response to eruptions? Or that there is too little internal, low-frequency variability (e.g., Laepple and Huybers)? Or does this suggest both, or something else?

Reply: The potential reasons for the stronger volcanic signal in some models compared with some reconstructions are varied and could include those stated by the referee alongside other reasons (notably errors and biases in the reconstructed temperatures, AMO, PDO and/or forcing histories). We would prefer not to over-speculate at this point and instead present the findings.

Page 10, line 2-3: the authors state that 'all models display red spectra'- in the methods (and in the time series in the figures), it seems that the data have been low-pass filtered, so by definition, the high-frequency variability has been reduced relative to the low-frequency variability (thus reddened)- I'm not sure that 'redness' really means

anything in this case. If 'redness' does mean something after the data have been fil-
tered, or if the data have not been low-pass filtered before spectral estimation, please
clarify/explain- for example, if the authors mean to say that one model has more lowfre-
quency variability than another, that may be more accurate.

Reply: Here, the spectral estimation is calculated without applying the low-pass filter
to the data. (i.e. the data have not been low pass filtered before spectral estimation).
We will ensure that this is clear in a revised manuscript.

Furthermore, the 'pronounced multidecadal variability' barely surpasses the 90% sig-
nificance threshold, as do most of the 'significant' peaks referenced in this paragraph.

Reply: As noted above, the presence of individual periodicities is of less interest than
the overall shape of the spectra (to repeat here for convenience: "overall shape of the
spectra, their redness and broad multi-decadal power, and whether these are simi-
lar between models, with/without forcing, and between AMO index definitions"). This
paragraph discusses some of these features and not individual significant periodici-
ties, so it is not affect by the choice of the significance threshold. We may remove the
significance lines to avoid this confusion.

These power spectra (AMO and PDO power spectra figures) are shown without any
error bars- when the spectra are compared and declared similar/different, some sort
of spectral estimation confidence bound/error bar on the figure could show if these
differences fall within the confidence bounds of the spectral estimates.

Reply: We will compute and add confidence intervals to the spectra.

Page 10, Lines _5-15: Perhaps this is the first time that this analysis has been done,
but I would be surprised- has anyone else compared the power spectra across these
simulations before? For example, Cheung et al., (2017) compares instrumental-based
AMO and Pacific variability to CMIP5 historical simulations (and also how the spatial
patterns associated with these modes can change through time). Parsons et al., 2017

(J Clim) compares instrumental, AR1, and CMIP5 Last Millennium, and CMIP5 Control spectra over the North Pacific and North Atlantic, and Fredriksen and Rypdal (2016, J Clim) compare spectra over ocean basins in CMIP5 models.

Reply: As per our understanding, we didn't find any study that compared the power spectra across the simulations in detail. As the reviewer mentioned, Parsons et al. (2017) discussed the power spectra in terms of ensemble mean of CMIP5 models but not the details of the power spectrum of individual CMIP5 models. Similarly, Cheung et al. (2017) did mention about the power spectrum of ensemble mean for the historical period and not the details of the individual members nor the last millennium runs. Fredriksen and Rypdal (2016) compared the power spectrum of CMIP5 control runs with instrumental records and not compared with last millennium simulations. So, we focused on the power spectra of individual CMIP5 models used in our study and compare the results between control and last millennium simulations.

Page 10, Line 23: the authors claim that the spatial patterns of AMO and PDO are similar to the patterns from observations. I see no comparisons among modelled and observed spatial patterns of variability. In fact, it would be helpful if the authors would show the spatial patterns from observations (of course acknowledging that the instrumental-based data have their own limitations) in Figures 1 and 2- this would help put the model results in context.

Reply: Now we have added the spatial patterns of AMO and PDO using observation data. See Figure R1 and R2.

Page 10, line 25: again, it's unclear if the data have been low-pass filtered before spectral analysis. Also, see my above comments- saying the spectra are 'red' seems meaningless if the data have been low-pass filtered. Again, the significant peaks barely surpass a 90% threshold- please discuss or mention if this significance is sensitive to threshold level.

Reply: The data have not been low pass filtered before the spectral analysis. We will

make this clear in the revised manuscript.

Also, as stated above it would be good to include error bars/lines on the spectra to know if the 'significant' differences from the background spectrum significant given uncertainties in the power spectral estimation?

Reply: We will compute and add confidence intervals to the spectra.

Page 11, _Line 25: good point.

Reply: Thank you.

Page 12, _line 4: OK, so other recent methods have been used to reconstruct SAT fields (e.g., Last Millennium Reanalysis from Hakim et al., 2016, JGRA and Tardiff et al., in review at CP)

Reply: We will cite the most recent Last Millennium Reanalysis paper and note that this does provide a surface temperature field that could be used to define an index based on the difference between the regional and global SST (though it is not independent of the climate model used to produce the reanalysis, so there may be some circularity in using the resultant AMO reconstruction to evaluate climate model behaviour).

Figures: Figures 1, 2, 3, 5: please include panels showing similar analyses from instrumental-based data products.

Reply: Now we have included panels based on the data from the instrumental period for Fig1 and 2. We have not included the instrumental data analysis for Figure 3 and 5, because the data length is not enough to calculate the correlation which is passed through 30-year low pass filtered.

Figure 4, Figure 6: it is interesting to see the PDO-E Asia and AMO-E Asia differences, but it would be nice to see some confidence bars on the control run values. For example, Coats et al. 2013 show that teleconnections can change from century to century. Could you do some sort of running correlation or subsample the control run to see how

variable this E Asian relationship is (or is there enough data?)

Reply: Confidence intervals could be included instead of the indicator of statistical significance (which occurs when the confidence interval does not include zero) and we will consider how best to graphically present this when we revise the figure (because it also needs revision to include additional GCM results). A running correlation or equivalent is not appropriate here because we are working with 30-year smoothed data (so that we can assess multi-decadal variability rather than the interannual variability that Coats et al., 2013, considered) and dividing it century by century would leave insufficient independent 30-year samples in each century.

Figure 9: Is there a way to put these results in context? For example, if you include the post-Pinatubo response in these models, could you show how the models compare to obs? Which models are more realistic? (CCSM4/MPI or BCC?)

Reply: The issue with BCC appears to be confined to the Last Millennium simulation and not to the Historical simulation, so a comparison with observations post-Pinatubo would not help.

Figures 10 and 11: inclusion of instrumental-based spectra could be helpful here too how realistic are these reconstructions?

Reply: We can include the instrumental based spectra although the data length is short.

Compact listing of purely technical corrections (typing errors, etc.). Page 1, Line 13: 'and their spectral characteristics'- remove 'their' Page 3, line17: change sentence to: 'Each model version was the same across all the simulations.' Page 4, line 8: 'much differences'- please re-word (e.g., 'A pattern correlation statistic shows minimal differences among : : :' Page 5, Line 7: 'largely suffer from the influence of external forcing' Page 6, Line 4: 'no time-varying (transient?) external radiative forcing' Page 6, line 31: 'This situation is equivalent to (that?) of Fig.' – there appears to be a

missing word here Page 7, Line 13-15: 'in the southern parts': : :'in all three models': : :'with the strongest correlation in the northeast region' Page 7, line17: 'though it varies' Page 9, Line 10: the sentence starting with 'Despite' appears a bit awkward- suggest rewording.

Reply: Thank you for the careful checking – we will address these minor technical/wording errors in our revised manuscript.

Please also note the supplement to this comment:
https://www.clim-past-discuss.net/cp-2018-164/cp-2018-164-AC2-supplement.pdf
* * *
[Figure]

**Fig. 1.** AMO SST patterns defined by regression of SST on the AMO index for each GCM, and for the observations (bottom row). Columns show results for different simulations (PI/LMH) and two definitions of AMO.

[Figure]

**Fig. 2.** PDO SST patterns defined by the first EOF of SST for each GCM, and for the observations (bottom row). Columns show results for different simulations (PI and LMH)

[Figure]

**Fig. 3.** Correlations between E Asian temperatures and AMO index for each GCM. Columns show results for different simulations (PI and LMH) and two definitions of the AMO index.

[Figure]

[Figure]

**Fig. 4.** Correlations between E Asian temperatures and PDO index for each GCM including the three additional models now analysed. Columns show results for different simulations (with/without forcings).

**Fig. 5.** As Fig. R3 but for the comparison between Last Millennium (850-1850) simulation (columns 1 and 2) and combined Last Millennium plus historical (850-2000) simulations (columns 3 and 4).

[Figure]

**Fig. 6.** As Fig. R4 but for the comparison between Last Millennium (850-1850) simulation (column 1) and combined Last Millennium plus historical (850-2000) simulations (column 2).

**Fig. 7.** Figure R7: Comparison of PDO pattern calculated from annual-mean SST (left) and monthly-mean SST (right).

---

## Author Response (AR1)

**"Identifying teleconnections and multidecadal variability of East Asian surface temperature during the last millennium in CMIP5 simulations" by Satyaban B. Ratna et al.**

We have responded to the referee's comments in blue text below, as well as modifying our manuscript and providing a new supplement containing some additional supporting results.

**Replies to anonymous Referee #1**

Ratna et al., examined the relationships between AMO/PDO and surface temperature in East Asia (TAS) at multidecadal time scales based on models and reconstructions data, found that external forcing greatly strengthened the relationship between AMO and TAS but weakened relationship between PDO and TAS, and discussed the volcano influences. This is an interesting study on how external forcing influences on teleconnetions between AMO/PDO and TAS. However, I still have some concerns on this study.

Reply: We thank the referee for their time and for their helpful suggestions.

Major concerns:

1) On the reliability of model and reconstruction data. Comparisons between modeled PDO/AMO from (CCSM4, MPI-ESM-P, BCC) with observed PDO/AMO index from HadISST/NCDC ERSST during the period of 1870-2000 should be added to evaluate the reliability of PDO/AMO index from model. There are several PDO/AMO reconstruction (such as, Gray et al., 2004; Shen et al., 2006). Although such PDO/AMO reconstructions are relatively short, the results seem more convincing by adding these records. In addition, there are published and robust Asian summer temperature reconstructions (e.g. Cook et al., 2013, Shi et al., 2015), such reconstruction data should be used. Comparisons among different reconstructions are as important as comparisons among the different models.

Shen, C., W.-C. Wang, W. Gong, and Z. Hao. 2006. A Pacific Decadal Oscillation record since 1470 AD reconstructed from proxy data of summer rainfall over eastern China. Geophysical Research Letters, vol. 33, L03702, 2006.
Gray, S.T., L.J. Graum-lich, J.L. Betancourt, and G.T. Pederson. 2004. A tree-ring based reconstruction of the Atlantic Multidecadal Oscillation since 1567 A.D. Geophysical Research Letters, 31:L12205, doi:10.1029/2004GL019932.
Cook E R , Krusic P J , Kevin J. Anchukaitis et al. Tree-ring reconstructed summer temperature nomalies for temperate East Asia since 800 C.E. Climate Dynamics, 2013, 41(11-12):2957-2972.
Shi F , Ge Q , Yang B , et al. A multi-proxy reconstruction of spatial and temporal variations in Asian summer temperatures over the last millennium. Climatic Change, 2015, 131(4):663-676.

Reply:

*Evaluation of model PDO/AMO.* We now include a spatial comparison of the models' PDO/AMO signatures with those observed PDO/AMO, using HadISST and ERSST in Figs. 1 and 2 of our revised manuscript.

*Analysis of additional, shorter PDO/AMO reconstructions.* The primary focus of this study is on the model simulations and the influence of external forcings, and on records of at least 1000-year length. While analysis of additional reconstructions is worth doing, that is more suited to another study examining shorter periods. We would prefer not to dilute our focus by analysis of additional reconstructions that are shorter (and, since our focus is also exclusively on multidecadal timescales and longer, having millennial-length timeseries is beneficial). Concerning the Gray et al. AMO reconstruction, we note the comments of Wang et al. (2017: disclosure, several authors are also authors of the current study): "The reconstruction of Gray et al. is based on a sparse tree-ring network, completely independent of our predictors; it has precise dating control, but its smaller network (only 12 sites) may compromise its representation of AMV if the centres of climate impact of AMV shift through time (also see the discussions in ref. 16)."

Wang J, Yang B, Ljungqvist FC, Luterbacher J, Osborn TJ, Briffa KR and Zorita E (2017) Internal and external forcing of multidecadal Atlantic climate variability over the past 1,200 years. Nature Geoscience 10, 512-517 (doi:10.1038/ngeo2962).

*Analysis of individual Asian summer temperature reconstructions.* The E Asian temperature reconstruction we used, from Wang et al. (2018), is a composite of seven published reconstructions that already *includes* the two suggested by the referee (Cook et al. 2013, Shi et al. 2015). The time series comparison of these three datasets can be seen in Wang et al (2018). Nevertheless, we have calculated the correlations between three of the individual summer temperature reconstructions (Cook et al. 2013, Shi et al. 2015, Zhang et al. 2018) and AMO (Wang et al. 2017, Mann et al. 2009), PDO (Mann et al. 2009, MacDonald et al. 2005), volcanic (GRA, Gao et al. 2008; CEA, Crowley et al. 2008; SIG, Sigl et al. 2015) and solar forcing (VSK, Vieira et al. 2011; DB, Delaygue and Bard (2011); SBF; Steinhilber et al. 2009). They do show some interesting differences, perhaps related to how well they resolve the response to volcanic forcing, so we have included these additional results in the supplementary material of our revised manuscript (Fig. S3). However, with the exception of the correlations between E Asian temperature and the Mann et al. (2009) AMO index, the only significant correlations (with some solar forcing, PDO and AMO reconstructions) at the multidecadal timescales are with the Wang et al. (2018) composite reconstruction.

2) On PDO signal. PDO has clear decadal and inter-decadal signal. Figure 11 also showed significant 15-20 years periods for PDO. However, all the time series are passed through a 30-year low pass filter using the Lanczos filter, which may miss key information of PDO. 10-year low pass filter should be used for PDO analysis.

Reply: We agree that PDO has a decadal signal which can been identified in Figure 11. However, the

specific purpose of our study is to look at variability on *multidecadal* timescales (title and first line of the abstract), not decadal timescales, which is why we have passed all timeseries (including the PDO) through a 30-year low-pass filter. Therefore we do not use a 10-year filter because that would conflict with the aim of our study.

3) On Volcano influences. Although previous studies showed that volcano eruptions affected decadal climate changes, it is equivocal that volcano eruptions affected multidecadal climate changes. For example, TAS reconstruction showed clear volcanic forcing signal, and volcano eruptions resulting in pulses of cooler summer conditions that may persist for several years (See Figure 12 in Cook et al., 2013). However, this study showed that there were not significant correlations between TAS and volcanic forcing (Figure 8c). In addition, superposed epoch analysis (SEA) should be used to test the impact of explosive volcanism on temperature.

Reply: Again, we note the aim of our study is explicitly to look at *multidecadal* timescales, so a superposed epoch analysis would not add more to the results already found. However, as noted above, we have now also analysed three individual E Asian temperature reconstructions, including Cook et al. (2013). Although the correlations with volcanic forcing are slightly stronger for Cook et al. (2013) than for the other reconstructions (Fig. S3 of our revised manuscript), they are still not statistically significant at the multidecadal timescale. This contrasts with six of the seven climate models, which show significant multidecadal correlations between simulated E Asian T and volcanic forcing, a finding that we report in the paper. There is also evidence that volcanic eruptions affect heat content and SST on these longer timescales (e.g. Gleckler et al. 2006). As suggested by the referee, the volcanic influence on the reconstructed temperatures is probably limited to the interannual timescale – but this timescale is not the focus of our study.

Gleckler, P. J., T. M. L. Wigley, B. D. Santer, J. M. Gregory, K. AchutaRao, and K. E. Taylor, 2006: Volcanoes and climate: Krakatoa's signature persists in the ocean. Nature, 439, 675.

4) On time scales of external forcing. there are other external forcings (e.g. solar activity) that should be considered. Solar activity has multi-decadal periods.

Reply: We focussed on volcanic forcing because it had previously been established that this was the largest external influence on the last millennium simulations (see the first paragraph of section 5 of our manuscript). However, since we are also looking at reconstructions, the referee is correct that we should not neglect solar forcing, because the reconstructions might show a significant association with solar forcing (indeed, as Wang et al. 2018 showed) even though the models may not. We have now included solar forcings in our analysis of the correlations between E Asian temperature and PDO/AMO/external forcing, both in model and reconstructions (Figs. 7 and 8 of our revised manuscript). We used three different solar forcing reconstructions (Vieira et al. 2011; DB, Delaygue and Bard (2011); Steinhilber et

al. 2009). The only E Asian temperature reconstruction with significant multidecadal correlations to solar forcing is the composite reconstruction of Wang et al. (2018).

5) On influences of external forcing, external forcing greatly strengthened the relationship between AMO and TAS but weakened relationship between PDO and TAS. Do you think such results are related to definition and calculation of AMO and PDO? In simple terms, AMO reflects average SST, but PDO reflects spatial configuration of SST. So AMO may be related to external forcing while PDO may be related to internal variability.

Reply: Yes, the referee's explanation is correct.

Minor Concerns:
1) Page 3, Line 20-22. For temperature over East Asia, TAS reconstruction is summer temperature, and TAS model data is summer, cold season temperature and annual temperature. It is confusing. Please clarify which season temperature used in Figure 3-8 in Figure caption.

Reply: The annual mean temperature is used in Figures 3-8. We have amended the figure captions to make this clear.

2) Page 13, Line 9, East Asiantemperature should be East Asian temperature.

Reply: We have corrected this typo.

3) Figure 7a, the label for y axis for volcanic forcing should be added.

Reply: We have added the y-axis label for volcanic forcing: 'Radiative forcing (W m$^{-2}$)'

4) Figure 8, confidence level explanation should be added.

Reply: We have changed the way that we mark the significant correlations (bold symbols or dashed lines for significance levels) and we have added to the captions that these indicated 'values significant at 95% level using a two-tailed student t-test'.

**Replies to Anonymous Referee #2**

General Comments: Ratna et al. examine the influence of transient external forcing (volcanic eruptions) on PDO and AMO variability and teleconnection patterns as they relate to East Asian surface air temperatures (SAT) in three PMIP3/CMIP5 past1000 simulations and paleoclimate reconstructions. This is an interesting study, and the results have interesting implications for how external forcing can impact internal variability and teleconnections. However, more work is needed to compare model output to observations and expand the study to other models.

Reply: We are grateful to the referee for their careful review and that they consider our work to be of interest. We respond below to the suggestions for expanding the scope of the work.

**Main Concerns:**

1) There are at least ten CMIP5/PMIP3 past1000 (Last Millennium) simulations available on ESGF that span the 850-1849 CE time period (BCC, CCSM4, CSIRO, FGOALS, GISS, Had, IPSL, MIROC, MPI, MRI). The authors exclude several of these simulations (MIROC, FGOALS, GISS) due to spin up/model drift/trend issues and cite Atwood et al for why they exclude these simulations. However, the authors choose not to use the output from CSIRO, HadCM3, IPSL, or MRI (some of which are included in the analysis of Atwood et al). The results therefore seem incomplete and selectively presented- why the exclusion of these other simulations? Please include analyses of these other Last Millennium simulations or at least provide a reason for why these other Last Millennium simulations have been excluded (the data have been available for at least 8-12 months online, so I hope it's not a data availability issue?). As the manuscript is currently written, 1/3 of the models show a completely different result, but this is only one model- is this really 1/3 of all CMIP5 Last Millennium models, or just one outlier in the CMIP5 Last Millennium simulations?

Reply: We originally considered all six models that had CMIP5/PMIP3 Last Millennium and CMIP5 historical simulations and that had data for all the necessary variables in the UK JASMIN facilities. We then discarded three due to drift issues as explained in our manuscript. Following the referee's recommendation, we sought data for the additional models suggested from other parts of the ESGF and we obtained sufficient data to extend our study to another four CMIP5 models (HadCM3, MRI, IPSL and CSIRO-Mk3L-1-2). We agree that our revised manuscript has been strengthened by including these additional model results. Some of the results are similar, but there are some differences in the correlations with E Asian temperature that we discuss in our revised manuscript.

2) The authors concatenate the Last Millennium (850-1849CE) and the Historical simulations (1850-2005CE) after removing the linear trend from each of these time segments separately. Removing a linear trend from either instrumental or CMIP5 data over the entire 1850-2005CE time period can be problematic if the main component of the 'warming trend' is in the 20th century. Multiple papers choose to remove the linear trend over the 20th century only (e.g., Deser et al., 2010; Messie and Chavez, 2011;

Franzke, 2014, Nature Climate Change; Ji et al., Nature Climate Change, 2014). Similarly, many CMIP5 historical simulations appear to show much of the global warming trend starting in the 20th century, so removing a trend over the full historical simulation period (1850-2005) may add in decadal-centennial variability. To avoid this detrending and concatenation problem, could the analysis just be conducted over the 850-1849CE time period (especially because it seems the authors are mostly focused on the impacts of volcanic eruptions on the PDO and AMO in the pre-1850CE time period?). Some recent work even suggests that the dynamics of the system change once GHG forcing becomes dominant (e.g., Song and Yu, 2015, J Clim; Brown et al., 2017, Nature Climate Change), so including this time period could be arguably problematic.

Reply: We recognise the referee's concerns about linearly detrending the Historical simulations, but our findings are not sensitive to this choice.

We note that we are trying to replicate in the models some aspects of what other studies have done using observations (proxy-based reconstructions and/or instrumental) and (a) in some cases linear detrending over the instrumental era is done even though it may not be optimal for the reasons given by the referee; and (b) the timing of the start of the anthropogenic warming can be in conflict with a linear detrending that begins in 1850 but this would be ameliorated for the PDO and for the AMOrem indices by the prior removal of global-mean SST from the Atlantic or Pacific SST values.

Nevertheless, we have tested to see whether our findings are sensitive to this issue by restricting the analysis to only the Last Millennium simulation and found that our results are quite similar to those we obtained by the combined detrended LM plus detrended historical simulations (compare columns of Figs. S1 and S2 for correlations with E Asia temperature for AMO and PDO, respectively).

We report this sensitivity test in the revised manuscript and discuss the few small differences that do occur. We keep the main results based on the combined LM+Historical analysis because the benefits of having a longer series to analyse outweighs the concerns raised now that we have shown that our findings are not sensitive to this issue.

3) There is no comparison between the spatial patterns of the AMO and PDO in instrumental-based reconstructions and the three models used here- perhaps some of these simulated spatial patterns are more realistic than others? The authors state that the model results are realistic, but never show this in the manuscript. The Climate Variability Diagnostics Package (http://webext.cgd.ucar.edu/Multi-Case/CVDP_ex/CMIP5-Historical/) shows that the spatial expressions of the AMO (and PDO) can be quite different in the various CMIP5 Historical simulations. Interpretation of the model results may be viewed through a more informed perspective if the models are compared to instrumental-based observations.

Reply: We now include a comparison of the AMO and PDO patterns with the instrumental data (similar comment from referee 1) in our revised Fig. 1 and 2.

4) Varying significance levels are used in the paper (90% vs 95%). Please use a consistent 95% or 99% significance level- as the paper stands, it appears that the significance level has been lowered to show 'significance spectral peaks' (e.g., Fig 10), but the spectra barely surpass this 90% level- why not use 95% or 99% everywhere? At least please include some discussion of the sensitivity of the results to significance level if the results don't pass this higher threshold (significance levels are admittedly is arbitrary, but the current, inconsistent use of 90% runs the risk of appearing selectively low to attempt to present a 'significant' result).

Reply: Significance tests are reported for three types of analysis in the manuscript. (1) For correlations between area-averaged temperature and driving factors (bar charts) we used the 'standard' 95% level. (2) For correlations between temperature fields and driving factors (contoured maps) we lowered this to the 90% level because the additional noise at the grid cell level increases the risk of a type II error (wrongly failing to reject the null hypothesis that there is no correlation). (3) For power spectra of AMO and PDO, we used a 90% level, but actually our interest is not really in the significance of the individual spectral peaks (and whether they pass an arbitrary level or not) but in the overall shape of the spectra, their redness and broad multi-decadal power, and whether these are similar between models, with/without forcing, and between AMO index definitions. We explain this better in the revised manuscript and we removed the significance lines from the spectra.

**General minor issues:**
Many authors abbreviate pre-industrial Control as PI (e.g., Atwood et al., 2016, J Clim, among others)- in an effort to maintain some sort of standard abbreviation that may be quickly recognized, I would encourage the authors to employ more commonly used acronyms (e.g., PI or piControl).

Reply: We modified the manuscript to use the abbreviation PI for pre-industrial Control.

Also, when reading through the figures, it is difficult to interpret the acronyms used in each figure without searching through the other figure captions or the text for the definitions of the acronyms- please define the acronyms used in each figure in each figure caption (or at least reference where they are defined) so readers can quickly understand the figure without searching for what they mean.

Reply: We now define the acronyms in the Figure captions.

**Specific comments:**
Page 1, Line 12-13: The simulated PDO and AMO spectra and spatial patterns are never compared to instrumental-based patterns or spectra (or even to proxy-based spatial patterns). Please include figures/analysis that support this statement in the main text or remove it.

Reply: We now include a comparison of the AMO and PDO patterns with the instrumental data (Fig. 1 and 2). We have also compared the spectrum of reconstruction and instrumental data for both AMO (Fig. 10) and PDO (Fig. 11).

5    Page 2, _line 10: The previous paragraph critiques the instrumental and proxy-based records, but little attention is paid to potential model deficiencies- can you at least briefly discuss or cite a few papers that may critique or even acknowledge that CMIP5/PMIP3 models have their own biases and problems as they relate to low-frequency SAT variability (e.g., Laepple and Huybers, 2014; Parsons et al., 2017 J Clim; ) or 'modes' of internal variability (or their responses to stratospheric aerosol loading from volcanic
10    eruptions)? Alternatively, directing the reader to where these model deficiencies, and their implications for your results, are going to be discussed later in the paper could be helpful.

Reply: we now cite these references and added a few sentences to describe their implications for potential model deficiencies. We discuss that Laepple and Huybers (2014) found potential deficiencies in CMIP5
15    SST variability, with model simulations diverging from a multiproxy estimate of SST variability (that is consistent between proxy types and with instrumental estimates) toward longer timescales. Parsons et al. (2017) found very different pictures of natural variability between CMIP5 models, including the North Atlantic, and between models and paleoclimate data in the tropics, in terms of the magnitude and spatial consistency of climate variance across interannual to centennial timescales.

Laepple, T., and P. H. Huybers, 2014: Ocean surface temperature variability: Large model–data differences at decadal and longer periods. Proc. Natl. Acad. Sci. USA, 111, 16 682–16 687, https://doi.org/10.1073/pnas.1412077111.

25    Page 3, lines 5-6: please see general comments in previous section- why were the bulk of the CMIP5/PMIP3 Last Millennium simulations excluded? Analysis of results would appear much more robust if an attempt is made to present more than 1/3 of the Last Millennium simulations, or if reasoning can be given why the other simulations were excluded. Also, what is the cutoff used for a drift that is 'too strong'? Is this a global or local drift? All the CMIP5/PMIP3 past1000 simulations appear to show some
30    sort of trend/drift at many grid points- the question is what is too much for the purposes of this AMO/PDO teleconnection study. Please clarify.

Reply: This comment has been addressed under the first "Main Concerns" earlier: we have extended our analysis to include four more models CSIRO, HadCM3, MRI and IPSL and revised the manuscript to
35    include these models and compare the additional results.

Our decision to exclude three CMIP5 models (MIROC-ESM, FGOALS-s2 and GISS) was based on the results discussed in Atwood et al. (2016), Fleming and Anchukaitis (2016) and Bothe et al. (2013). This is mentioned in our original manuscript. Atwood et al (2016) and Bothe et al (2013) discussed long-term
40    drift in global mean surface air temperature in their PI simulations. Similarly, Fleming and Anchukaitis

(2016) found drift in the Last Millennium simulations, which are apparent in the initial several centuries and excluded from their PDO analysis.

Page 3, Line 20: please see general comments in previous section- removing one linear trend over the full 1850-2005CE time period seems like it may add in low-frequency variability, and I am still not even sure why the historical simulations have been included if the focus is on the impact of volcanic eruptions in the pre-historical simulation time period.

Reply: This comment has been addressed under the second "Main Concerns" earlier. Our findings are not sensitive to this choice and we explain this in the revised manuscript. Our focus is broader than just the impact of volcanic eruptions, we are interested in the influence of external forcings in general on the diagnosis of the role of internal variability from observational evidence. During these simulations, volcanic forcing plays a major role so we did some additional analysis of that but it is not our only result.

Page 4, Line 8: 'we don't see much differences'- this is a subjective statement. What criteria are used? Perhaps something like a pattern metric or Euclidean distances metric could be used to say something more quantitative?

Reply: we revised this sentence to be less subjective, noting which key features (position and strength of the loading maxima and loading gradients) of the PDO patterns are present in the simulated and observed fields.

Page 4, Line 10: Please explain how the TAS time series is made- I assume annual mean (Jan-Dec?) temperature at each grid box, latitude-weighted, and masked ocean grid boxes? Over what latitude and longitude range is this area average made (is it the whole region used in the maps in the figures showing East Asia?)? Please provide more details in the text.

Reply: For the TAS time series, the annual mean (Jan-Dec) TAS is calculated over the land grid points only and area averaged over the region 60E -150E and 10N-55N. We added this additional information to the revised manuscript.

Page 5, line 5-7: There are other PDO reconstructions- fine to not include them, but can you state why this one is selected over others?

Reply: We have used the selected PDO reconstructions based on the availability of the data for a longer period that covers at least 1000 years of our main analysis period 850-2000. The revised manuscript now states this selection criterion.

Page 5, lines 11-15: As far as I can tell, the model-based PDO indices are made from monthly data, and the paleo-based PDO indices are made 'annual' data (or seasonally sensitive proxy records)- would a

better comparison be to make annual means of SAT for the model data, then construct the PDO index, so the index is more comparable to the annual proxy-based index? (or can you show that the annual and monthly modelbased PDO patterns and time series are similar?).

5 Reply: The model simulated monthly mean SST data were, in fact, already converted to annual mean data before applying the EOF analysis to get the PDO pattern and its time series (i.e. as suggested by the reviewer – we now make this clear by adding 'annual-mean SST anomalies' to the Fig. 2 caption). We have done this because all our analysis is based on the annual mean data, which also compares with the annual mean reconstructed data. We have also confirmed that model based PDO patterns for annual and
10 monthly data are similar (see figure R7 of our reply in the interactive discussion).

Page 5, Line 21, line 25: The 90% significance level seems oddly low, and arbitrarily used in only certain cases- do your results consistently pass a 95% significance test (both the regions in the maps and the spectra)? For example, the 'significant' spectral peaks in Figure 10 appear quite close to the 90%
15 significance level- if you made this a 95 or 99%, are these 'significant spectral peaks' at all significant?

Reply: Please see our response to "main concern (4)" above.

Page 8, line 26: the authors discuss a weak response in BCC to volcanic eruptions- is this a finding that
20 has been noted previously (e.g., Driscoll et al., JGR, 2012, or some sort of similar CMIP5 comparison to observations?)? How realistic is this model's response relative to the other models' responses to volcanic eruptions (especially compared to observations of more recent eruptions and their impacts)? I ask because this difference seems to be important to the results- for example, should the BCC changes (or lack thereof relative to the other models) in PDO, AMO, and associated teleconnections with E Asia be viewed as just
25 as realistic as the other models' responses? Or is it an outlier because it doesn't respond at all to volcanic eruptions when it should?

Reply: By analysing the CMIP5 *historical* simulations, Driscoll et al. (2012) found largest anomaly in the reflected SW radiation in the BCC model. Here, we show that the weak response in BCC to volcanic
30 response only exists in the *last millennium* simulations, where we have analysed three major volcanic eruptions that happened in the last millennium. We also analysed the same for the major volcanic events during the historical period but didn't find such weak response in BCC model compared to the other models. So, it seems that the weak volcanic forcing and response in BCC GCM only exist in its last millennium simulation.
35
Page 9, Line 25-26: It would be helpful to show results from the other four CMIP5 Last Millennium simulations here to put these results in context- right now, 1/3 of the models show a completely different result, but this '1/3 of models' is just the BCC model.

40 Reply: As noted earlier, we now analyse four other models and it has improved our manuscript.

Page 9, Line 26-29: Would this result imply that the models show an unrealistically large response to eruptions? Or that there is too little internal, low-frequency variability (e.g., Laepple and Huybers)? Or does this suggest both, or something else?

Reply: The potential reasons for the stronger volcanic signal in some models compared with some reconstructions are varied and could include those stated by the referee alongside other reasons (notably errors and biases in the reconstructed temperatures, AMO, PDO and/or forcing histories). We prefer not to over-speculate at this point and instead present the findings.

Page 10, line 2-3: the authors state that 'all models display red spectra'- in the methods (and in the time series in the figures), it seems that the data have been low-pass filtered, so by definition, the high-frequency variability has been reduced relative to the low-frequency variability (thus reddened)- I'm not sure that 'redness' really means anything in this case. If 'redness' does mean something after the data have been filtered, or if the data have not been low-pass filtered before spectral estimation, please clarify/explain- for example, if the authors mean to say that one model has more lowfrequency variability than another, that may be more accurate.

Reply: The data have **not** been low pass filtered before spectral estimation. We have now clearly mentioned this in the revised manuscript.

Furthermore, the 'pronounced multidecadal variability' barely surpasses the 90% significance threshold, as do most of the 'significant' peaks referenced in this paragraph.

Reply: As noted above, the presence of individual periodicities is of less interest than the overall shape of the spectra (to repeat here for convenience, we are interested in: "the overall shape of the spectra, their redness and broad multi-decadal power, and whether these are similar between models, with/without forcing, and between AMO index definitions"). This paragraph discusses some of these features and not individual significant periodicities, so it is not affected by the choice of the significance threshold. We have removed the significance lines to avoid this confusion.

These power spectra (AMO and PDO power spectra figures) are shown without any error bars- when the spectra are compared and declared similar/different, some sort of spectral estimation confidence bound/error bar on the figure could show if these differences fall within the confidence bounds of the spectral estimates.

Reply: We cannot add individual confidence intervals to each individual spectrum, especially now that there are seven GCMs, without obscuring the message of the diagram by two many lines. Using a log-scale for the y-axis would mean that a single confidence interval could be marked that applies to all frequencies but we decided not to do this because (a) a single confidence interval would only apply to all series if they are based on the same length timeseries (which is not true for the PI runs, though it is for

the LMH runs) and (b) we tried using a log-scale and felt that it made it harder to see the differences between the GCMs.

Page 10, Lines _5-15: Perhaps this is the first time that this analysis has been done, but I would be surprised- has anyone else compared the power spectra across these simulations before? For example, Cheung et al., (2017) compares instrumental-based AMO and Pacific variability to CMIP5 historical simulations (and also how the spatial patterns associated with these modes can change through time). Parsons et al., 2017 (J Clim) compares instrumental, AR1, and CMIP5 Last Millennium, and CMIP5 Control spectra over the North Pacific and North Atlantic, and Fredriksen and Rypdal (2016, J Clim) compare spectra over ocean basins in CMIP5 models.

Reply: As per our understanding, we didn't find any study that compared the power spectra across the simulations in detail. As the reviewer mentioned, Parsons et al. (2017) discussed the power spectra in terms of ensemble mean of CMIP5 models but not the details of the power spectrum of individual CMIP5 models. Similarly, Cheung et al. (2017) did mention the power spectrum of ensemble mean for the historical period and not the details of the individual members nor the last millennium runs. Fredriksen and Rypdal (2016) compared the power spectrum of CMIP5 control runs with instrumental records but did not compare with last millennium simulations. So, we focused on the power spectra of individual CMIP5 models used in our study and compare the results between control and last millennium simulations.

Page 10, Line 23: the authors claim that the spatial patterns of AMO and PDO are similar to the patterns from observations. I see no comparisons among modelled and observed spatial patterns of variability. In fact, it would be helpful if the authors would show the spatial patterns from observations (of course acknowledging that the instrumental-based data have their own limitations) in Figures 1 and 2- this would help put the model results in context.

Reply: We have added the spatial patterns of AMO and PDO using observation data (Fig. 1 and 2).

Page 10, line 25: again, it's unclear if the data have been low-pass filtered before spectral analysis. Also, see my above comments- saying the spectra are 'red' seems meaningless if the data have been low-pass filtered. Again, the significant peaks barely surpass a 90% threshold- please discuss or mention if this significance is sensitive to threshold level.

Reply: The data have not been low pass filtered before the spectral analysis. We make this clear in the revised manuscript.

Also, as stated above it would be good to include error bars/lines on the spectra to know if the 'significant' differences from the background spectrum significant given uncertainties in the power spectral estimation?

Reply: See earlier response.

Page 11, _Line 25: good point.

Reply: Thank you.

Page 12, _line 4: OK, so other recent methods have been used to reconstruct SAT fields (e.g., Last Millennium Reanalysis from Hakim et al., 2016, JGRA and Tardiff et al., in review at CP)

Reply: We will cite the most recent Last Millennium Reanalysis paper and note that this does provide a surface temperature field that could be used to define an index based on the difference between the regional and global SST (though it is not independent of the climate model used to produce the reanalysis, so there may be some circularity in using the resultant AMO reconstruction to evaluate climate model behaviour).

**Figures:**
Figures 1, 2, 3, 5: please include panels showing similar analyses from instrumental-based data products.

Reply: We have now included panels based on the data from the instrumental period for Fig1 and 2. We have not included the instrumental data analysis for Figure 3 and 5, because the data length is not enough to calculate the correlation which is based on 30-year low pass filtered data.

Figure 4, Figure 6: it is interesting to see the PDO-E Asia and AMO-E Asia differences, but it would be nice to see some confidence bars on the control run values. For example, Coats et al. 2013 show that teleconnections can change from century to century. Could you do some sort of running correlation or subsample the control run to see how variable this E Asian relationship is (or is there enough data?)

Reply: We considered using confidence intervals instead indicating the statistical significance (which occurs when the confidence interval does not include zero) but now that we extended our analysis to seven GCMs it is problematic to fit all the information without obscuring the individual model results. A running correlation or equivalent is not appropriate here because we are working with 30-year smoothed data (so that we can assess multi-decadal variability rather than the interannual variability that Coats et al., 2013, considered) and dividing it century by century would leave insufficient independent 30-year samples in each century.

Figure 9: Is there a way to put these results in context? For example, if you include the post-Pinatubo response in these models, could you show how the models compare to obs? Which models are more realistic? (CCSM4/MPI or BCC?)

Reply: The issue with BCC appears to be confined to the Last Millennium simulation and not to the Historical simulation, so a comparison with observations post-Pinatubo would not help.

Figures 10 and 11: inclusion of instrumental-based spectra could be helpful here too how realistic are these reconstructions?

Reply: We have included the instrumental based spectra although the data length is short.

**Compact listing of purely technical corrections (typing errors, etc.).**
Page 1, Line 13: 'and their spectral characteristics'- remove 'their'
Page 3, line17: change sentence to: 'Each model version was the same across all the simulations.'
Page 4, line 8: 'much differences'- please re-word (e.g., 'A pattern correlation statistic shows minimal differences among : : :'
Page 5, Line 7: 'largely suffer from the influence of external forcing'
Page 6, Line 4: 'no time-varying (transient?) external radiative forcing'
Page 6, line 31: 'This situation is equivalent to (that?) of Fig.' – there appears to be a missing word here
Page 7, Line 13-15: 'in the southern parts': : :'in all three models': : :'with the strongest correlation in the northeast region'
Page 7, line17: 'though it varies'
Page 9, Line 10: the sentence starting with 'Despite' appears a bit awkward- suggest rewording.

Reply: Thank you for the careful checking – we have addressed these minor technical/wording errors in our revised manuscript.

[revised manuscript text omitted]

---

## Referee Report (RR1)

cp-2018-164
Submitted on 26 Nov 2018

Identifying teleconnections and multidecadal variability of East Asian surface temperature during the last millennium in CMIP5 simulations

Ratna et al.

**Suggested minor changes listed by page in revised PDF manuscript:**

**Page 1**
Line 23: 'when interpreting internal variability teleconnections'- this phrase is long and confusing- for clarity, please divide up and clarify
Line 27: 'adjacent' continents- suggest 'both nearby and remove regions'
Line 30: 'each of which' – this implies you have already listed the methods

**Page 2**
Lines 13-15: the wording seems to imply that the models mentioned in previous lines were uncoupled- but the CMIP5 past1000 models used in previous studies are coupled, correct?

**Page 3**
Line 7: suggest change to 'time-varying external forcings'
Line 10: 'some PI simulations' and 'some LM simulation'- this wording sounds vague- please be more specific
Line 17: portion of variability in what? Please be more specific
Line 20: 'the all the'

**Page 4**
Lines 12-13 and 19-20: IPCC AR5 WG1 Chapter 9 summarizes AMO and PDO observation-model comparisons. May be helpful for the reader to cite this summary.

Flato, G., J. Marotzke, B. Abiodun, P. Braconnot, S.C. Chou, W. Collins, P. Cox, F. Driouech, S. Emori, V. Eyring, C. Forest, P. Gleckler, E. Guilyardi, C. Jakob, V. Kattsov, C. Reason and M. Rummukainen, 2013: Evaluation of Climate Models. In: Climate Change 2013: The Physical Science Basis. Contribution of Working Group I to the Fifth Assessment Report of the Intergovernmental Panel on Climate Change [Stocker, T.F., D. Qin, G.-K. Plattner, M. Tignor, S.K. Allen, J. Boschung, A. Nauels, Y. Xia, V.

Line 14: Please cite a paper related to how PDO is calculated so the reader can find more information if they need to replicate analysis, etc. (I assume you didn't come up with this method)

**Page 9**
Line 18: 'Turning to correlations some key behaviours are clear.' This sentence sounds vague – can you please clarify behaviours and be more specific with your wording?

**Page 10**
Line 23: I think 'their being' should be 'there being'

**Page 11**
Line 11: A mention of potential lagged relationships between modes of variability and volcanic eruptions is an important factor to acknowledge- for example, Pausata et al. (2015, PNAS) show that volcanic eruptions can impact AMOC variability decades after the eruption. The zero-lag regressions won't account for this kind of impact on the climate system.
Line 14: 'This is especially for two models'- This what? Please be more specific.

**Page 12:**
*Much of this section reads like it hasn't been proofread or hasn't been fully been updated after the first round of revisions (wording that still references three models, also wording that still references background red spectrum on figures, which has been removed post-revision)- I tried catching the most obvious problems, but please carefully read over this section to be sure it matches the figures and data presented.*

Line 4-5: the text mentions 'elevated above the background red spectrum'- there is no background red spectrum shown on Figure 10a. Please change wording or add in red spectrum.
Line 5: '(CCSM4 has elevated' never has a closing parenthesis
Line 9: Why are the authors only referring to three models? Are these the three models referred to at the end of the last paragraph, or have the authors not updated the text to discuss all 7 models? Not sure which.
Lines 10-11: 'Landrum et al found the same frequency for both control and last millennium simulations'. Unless the reader knows this paper, it is not clear what model this refers to. Please be more specific.
Line 12: 'weaken the PDO-TAS correlation'- In the CCSM4? All models? In the reconstructions? please be more specific.
Lines 14-15: 'The instrumental record…forcings at multidecadal timescales'- If a reader skips to this section and starts reading, they will have no frame of reference what you are talking about here- does this statement apply to all aspects of climate? The PDO? AMO? E Asian temps? please be specific.
Lines 23-24: Please reference your figure here for the reader. Also, among other papers, Parsons et al. 2017 (J Clim, Figure 6) show spectra of North Pacific and North Atlantic SST/SAT variability in CMIP5 piControl, CMIP5 Last Millennium, and instrumental-based data. Fleming and Anchukaitis (Clim. Dyn, 2016, Fig. 5) also show power spectra of the PDO time series. I recognize you cite these papers earlier, but it would be good to acknowledge that others have shown the power spectra of these 'modes' of variability in these CMIP5 models before.

**Page 13**
Lines 1-4: Great points here
Line 9: please be specific (e.g., spell out what effect is, or 'the effect mentioned in point (4) above')
Line 11: What does 'much but not all' mean? Please be more specific

Line 18: "This partly arises"- which part about what you were discussing? Confusing to follow- please be more specific.
Lines 22-27: Great points
Line 28: "there are a number of ways to attempt this"- please be more specific what 'this' means

**Page 14**
Line 21: 'determine its' – typo? This sentence doesn't make sense.

---

## Author Response (AR2)

To,
The Editor
Climate of the Past

Dear Editor,
We herewith submit a revised version of our manuscript entitled "Identifying teleconnections and multidecadal variability of East Asian surface temperature during the last millennium in CMIP5 simulations", for the Climate of the Past journal. We followed the referees' valuable comments, addressed each comment in the main text and responded to them point-by-point below. As required by you, please find attached here the following documents.

(i)     Revised manuscript
(ii)    Point-by-point reply to the comments
(iii)   Marked-up manuscript version

Thanking you.

Sincerely,

Satyaban Bishoyi Ratna
(On behalf of all co-authors)

Climatic Research Unit,
School of Environmental Sciences,
University of East Anglia,
Norwich,  NR4 7TJ, UK.
E-mail: s.bishoyi-ratna@uea.ac.uk
Tel: +44 (0)1603 59 3808

**"Identifying teleconnections and multidecadal variability of East Asian surface temperature during the last millennium in CMIP5 simulations" by Satyaban B. Ratna et al.**

We have responded to the referee's comments in blue text below, as well as modifying our manuscript.

**Replies to anonymous Referee #1**

Ratna et al. show significant improvement in the manuscript. They present analysis of more CMIP5 past1000 simulations, include a model-instrumental data comparison, and show improved explanations of their methods and results.

Most of my suggested changes (attached PDF) relate to minor wording issues and a few citation suggestions.

As I re-read the manuscript post-revisions, it occurred to me that the authors do not discuss the lagged effects of volcanic eruptions. The method of 'regressing out' volcanic eruptions appears to rely on instantaneous relationships between air temperature and volcanic aerosol forcing. However, a growing body of work (e.g., Pausata et al., 2015, PNAS, among others) shows that in CMIP5-class model simulations, there are significant lagged effects in the climate system (e.g., AMOC, sea ice, and ENSO changes) years to decades after a volcanic eruption. It seems the manuscript would be strengthened if this point is addressed somewhere in the manuscript.

I have noted several sections in the attached PDF where minor, technical wording issues could be corrected.

Reply: We are grateful to the referee for their careful review and helpful suggestions.

We agree with the concern of the reviewer that volcanic eruptions may have lagged effects in the climate system from years to decades. Since we are regressing out the volcanic signal using 30-year smoothed data, a short lag of 1 or 2 years makes little difference, but a longer lag (say 1 or 2 decades) would be of interest. The current evidence for a significant decadally-lagged **AMOC** response to volcanic forcing is strong in *some* CMIP5-class models (e.g. Pausata et al. 2015) but much *weaker* for the **AMO** response (e.g. Fig. 3 of Mignot et al. 2011 shows maximum N Atlantic SST response one year after eruption; Figs. 2d and 2e of Ottera et al. 2010 shows maximum AMO response 1 or 2 years after solar and volcanic forcing). Since our focus is on the AMO and E Asian temperature, the evidence for a decadally-lagged response is not sufficiently strong to change our approach. Nevertheless it is a relevant point, so we have now discussed/referenced this point in the revised manuscript.

Mignot et al. (2011) https://www.clim-past.net/7/1439/2011/
Otter et al. (2010) http://www.nature.com/doifinder/10.1038/ngeo955

**Page 1**
Line 23: 'when interpreting internal variability teleconnections'- this phrase is long and confusing- for clarity, please divide up and clarify
Reply: We have modified the sentence for clarity.

Line 27: 'adjacent' continents- suggest 'both nearby and remove regions'
Reply: Done.

Line 30: 'each of which' – this implies you have already listed the methods
Reply: Modified.

**Page 2**
Lines 13-15: the wording seems to imply that the models mentioned in previous lines were uncoupled- but the CMIP5 past1000 models used in previous studies are coupled, correct?
Reply: Correct. We have modified the sentence accordingly.

**Page 3**
Line 7: suggest change to 'time-varying external forcings'
Reply: Changed

Line 10: 'some PI simulations' and 'some LM simulation'- this wording sounds vague- please be more specific
Reply: Now we have specifically named the PI and LM simulations

Line 17: portion of variability in what? Please be more specific
Reply: We have now modified the sentence for clarity.

Line 20: 'the all the'
Reply: Corrected.

**Page 4**
Lines 12-13 and 19-20: IPCC AR5 WG1 Chapter 9 summarizes AMO and PDO observationmodel comparisons. May be helpful for the reader to cite this summary.

Flato, G., J. Marotzke, B. Abiodun, P. Braconnot, S.C. Chou, W. Collins, P. Cox, F. Driouech, S. Emori, V. Eyring, C. Forest, P. Gleckler, E. Guilyardi, C. Jakob, V. Kattsov, C. Reason and M. Rummukainen, 2013: Evaluation of Climate Models. In: Climate Change 2013: The Physical Science Basis. Contribution of Working Group I to the Fifth Assessment Report of the Intergovernmental Panel on Climate Change [Stocker, T.F., D. Qin, G.-K. Plattner, M. Tignor,

S.K. Allen, J. Boschung, A. Nauels, Y. Xia, V.
Reply: We have now cited this report as suggested.

Line 14: Please cite a paper related to how PDO is calculated so the reader can find more information if they need to replicate analysis, etc. (I assume you didn't come up with this method)
Reply: We have now cited a few papers which calculated the PDO index and our study is also similar to the methods defined by them.

**Page 9**
Line 18: 'Turning to correlations some key behaviours are clear.' This sentence sounds vague - can you please clarify behaviours and be more specific with your wording
Reply: We have now modified this sentence as the details of the correlation are discussed in the next paragraphs.

**Page 10**
Line 23: I think 'their being' should be 'there being'
Reply: Corrected.

**Page 11**
Line 11: A mention of potential lagged relationships between modes of variability and volcanic eruptions is an important factor to acknowledge- for example, Pausata et al. (2015, PNAS) show that volcanic eruptions can impact AMOC variability decades after the eruption. The zero-lag regressions won't account for this kind of impact on the climate system.
Reply: We agree with the concern of reviewer that volcanic eruptions may have lagged effects in the climate system from years to decades (e.g. Pausata et al. 2015). We have now addressed this point in the revised manuscript (and see our response above for more details).

Line 14: 'This is especially for two models'- This what? Please be more specific.
Reply: This sentence is modified now.

**Page 12:**
*Much of this section reads like it hasn't been proofread or hasn't been fully been updated after the first round of revisions (wording that still references three models, also wording that still references background red spectrum on figures, which has been removed post-revision)- I tried catching the most obvious problems, but please carefully read over this section to be sure it matches the figures and data presented.*
Reply: Thanks for these comments. We have now carefully looked at this section and made necessary changes, including the number of models now analysed.

Line 4-5: the text mentions 'elevated above the background red spectrum'- there is no background red spectrum shown on Figure 10a. Please change wording or add in red spectrum.
Reply: We have changed the wording. Each model spectrum is shown and they are mostly red, and we are referring to whether there is enhanced power at 60-80 years relative to the rest of the spectrum for that particular model.

Line 5: '(CCSM4 has elevated' never has a closing parenthesis
Reply: Corrected.

Line 9: Why are the authors only referring to three models? Are these the three models referred to at the end of the last paragraph, or have the authors not updated the text to discuss all 7 models? Not sure which.

Reply: Corrected.

Lines 10-11: 'Landrum et al found the same frequency for both control and last millennium simulations'. Unless the reader knows this paper, it is not clear what model this refers to. Please be more specific.
Reply: Now we have clearly mentioned by naming the model (CCSM4) they have used.

Line 12: 'weaken the PDO-TAS correlation'- In the CCSM4? All models? In the reconstructions? please be more specific.
Reply: Now we made it clear.

Lines 14-15: 'The instrumental record…forcings at multidecadal timescales'- If a reader skips to this section and starts reading, they will have no frame of reference what you are talking about here- does this statement apply to all aspects of climate? The PDO? AMO? E Asian temps? please be specific.
Reply: We have added an introductory sentence to the discussion and conclusions section to provide the frame of reference and which specifically mentions AMO, PDO and East Asian climate.

Lines 23-24: Please reference your figure here for the reader. Also, among other papers, Parsons et al. 2017 (J Clim, Figure 6) show spectra of North Pacific and North Atlantic SST/SAT variability in CMIP5 piControl, CMIP5 Last Millennium, and instrumental-based data. Fleming and Anchukaitis (Clim. Dyn, 2016, Fig. 5) also show power spectra of the PDO time series. I recognize you cite these papers earlier, but it would be good to acknowledge that others have shown the power spectra of these 'modes' of variability in these CMIP5 models before.
Reply: We have added a reference to the figures and now acknowledged these papers as suggested.

**Page 13**
Lines 1-4: Great points here
Reply: Thank you.

Line 9: please be specific (e.g., spell out what effect is, or 'the effect mentioned in point (4) above')
Reply: Now we have clearly mentioned this.

Line 11: What does 'much but not all' mean? Please be more specific
Reply: We have modified this paragraph to improve the clarity of this point.

Line 18: "This partly arises"- which part about what you were discussing? Confusing to followplease be more specific.
Reply: We have joined these sentences for clarity.

Lines 22-27: Great points
Reply: Thank you.

Line 28: "there are a number of ways to attempt this"- please be more specific what 'this' means
Reply: We have modified this sentence.

**Page 14**
Line 21: 'determine its' – typo? This sentence doesn't make sense.
Reply: We have modified this sentence to make it understandable.

**Replies to anonymous Referee #2**

Ratna et al., have addressed comments and improved the manuscript. Although it is difficult to distinguish the effects of external forcing and internal variability on the climate, the study has yielded some interesting results. The results show that the relationship between AMO\PDO and East Asian temperature depends on models, external forcing and definition of index. Such research will be helpful for future research based on climate simulations and proxy-based spatial SST reconstruction.

Reply: We thank the referee for their time and helpful suggestions.

A few minor comments:
1. P4 Line 29; Gao et al., (2008); Line 30 and Table 1, GRA=Gao et al., (2008), please revise GEA/CER to GRA/CEA in Table 1.
Reply: Corrected.

2. P12 Line 17, "seven" climate models.
Reply: Corrected

3. P14 Line 9, "because".
Reply: Corrected

4. Adding confidence level for Figure 10 and 11.
Reply: We have already mentioned why we didn't use confidence level by responding to one of the referee comments during the first revision. We cannot add individual confidence intervals to each individual spectrum, especially now that there are seven GCMs, without obscuring the message of the diagram by too many lines. Using a log-scale for the y-axis would mean that a single confidence interval could be marked that applies to all frequencies but we decided not to do this because (a) a single confidence interval would still only apply to all series if they are based on the same length timeseries (which is not true for the PI runs, though it is for the LMH runs) and (b) we tried using a log-scale and felt that it made it harder to see the differences between the GCMs.

[revised manuscript text omitted]